# Nucleoporin downregulation modulates progenitor differentiation independent of nuclear pore numbers

Amy E. Neely[1], Yang Zhang [2,5✉], Laura A. Blumensaadt[1], Hongjing Mao[5], Benjamin Brenner[2], Cheng Sun [3], Hao F. Zhang [2] & Xiaomin Bao [1,4✉]

Nucleoporins (NUPs) comprise nuclear pore complexes, gateways for nucleocytoplasmic transport. As primary human keratinocytes switch from the progenitor state towards differentiation, most NUPs are strongly downregulated, with NUP93 being the most downregulated NUP in this process. To determine if this NUP downregulation is accompanied by a reduction in nuclear pore numbers, we leveraged Stochastic Optical Reconstruction Microscopy. No significant changes in nuclear pore numbers were detected using three independent NUP antibodies; however, NUP reduction in other subcellular compartments such as the cytoplasm was identified. To investigate how NUP reduction influences keratinocyte differentiation, we knocked down NUP93 in keratinocytes in the progenitor-state culture condition. NUP93 knockdown diminished keratinocytes' clonogenicity and epidermal regenerative capacity, without drastically affecting nuclear pore numbers or permeability. Using transcriptome profiling, we identified that NUP93 knockdown induces differentiation genes related to both mechanical and immune barrier functions, including the activation of known NF-κB target genes. Consistently, keratinocytes with NUP93 knockdown exhibited increased nuclear localization of the NF-κB p65/p50 transcription factors, and increased NF-κB reporter activity. Taken together, these findings highlight the gene regulatory roles contributed by differential NUP expression levels in keratinocyte differentiation, independent of nuclear pore numbers.

[1] Department of Molecular Biosciences, Northwestern University, Evanston, IL 60208, USA. [2] Department of Biomedical Engineering, Northwestern University, Evanston, IL 60208, USA. [3] Department of Mechanical Engineering, Northwestern University, Evanston, IL 60208, USA. [4] Department of Dermatology, Northwestern University, Chicago, IL 60611, USA. [5] Present address: Molecular Analytics and Photonics (MAP) Lab, Department of Textile Engineering, Chemistry and Science, North Carolina State University, Raleigh, NC 27606, USA. ✉email: yang.zhangfl@gmail.com; xiaomin.bao@northwestern.edu

Nucleoporins (NUPs), a family of about 30 different proteins, are generally considered as the building blocks of nuclear pore complexes (NPCs). Nuclear pores perforate the nuclear envelope and serve as essential gateways for nucleocytoplasmic transport, permitting regulators to enter the nucleus from the cytoplasm for modulating gene expression. The composition and structure of the nuclear pores are surprisingly flexible in biological processes. Diversified NUP compositions evolved in different eukaryotes for their functional adaptation[1,2]; differential NUP stoichiometry was reported in NPCs isolated from different cell types[3,4]; functional deterioration of NUP occurs aging[1,2]. Moreover, several NUPs have been identified to function beyond the nuclear pores, including chromatin binding and ER-stress modulation[3–5]. These findings highlight the complexity of NUPs across different systems. Despite recent advances in understanding the structure of nuclear pores at near-atomic resolution in vitro[6,7], the function of NUPs in different cellular processes remain incompletely understood.

Around 300 billion cells are turned over in an average adult human body per day[8]. This cellular turnover predominately occurs in the self-renewing somatic tissues, such as blood and the epithelial tissues that line the surfaces and cavities of all major organs[9,10]. To compensate for the cell turnover while sustaining tissue function, a subset of the progenitors (adult stem cells) needs to continuously switch their state towards terminal differentiation. Dysregulation of the differentiation process underlies a spectrum of human diseases including cancer, and 90% of life-threatening cancers are originated from epithelial tissues. The cell-state transition from progenitors towards differentiation is a highly complex process, involving gene expression changes from thousands of genes to cease proliferation and progressively activate terminal differentiation[11,12]. Regulatory mechanisms at multiple stages of gene expression are involved in differentiation regulation, including signaling, transcription, RNA processing, and translation[11,13–17]. How NPCs participate in this cell state switch, from the progenitor state towards differentiation, remains under-characterized.

The skin epidermis, functioning as the first-line physical and immune barrier for the human body, is among the self-renewing somatic tissues that continuously regenerate throughout the lifetime. As a research platform, the skin epidermis is well-suited for investigating molecular mechanisms underlying progenitor differentiation. Primary epidermal keratinocytes can be isolated and expanded using well established protocols[18]. These progenitor-state keratinocytes can be switched to the differentiation state in vitro, progressively transitioning from early, mid, to terminal differentiation within the course of 6 days[12,19]. These progenitor-state keratinocytes can further be genetically modified and regenerate three-dimensional full-thickness epidermal tissue in organotypic culture systems[13,20,21]. Thus, this research platform allows the integration of biochemical, imaging, genomic, and genetic approaches to decipher molecular mechanisms.

As keratinocytes switch from the progenitor state toward differentiation, we find that most NUPs are strongly downregulated in this process, with NUP93 being one of the most downregulated NUPs. To determine if this NUP downregulation results in reduced nuclear pore numbers, we leveraged super-resolution imaging. Combining Stochastic Optical Reconstruction Microscopy (STORM) together with biochemical isolation of the nuclei, we optimized super-resolution imaging (at 25 nm resolution) with labeling of endogenous NUPs in primary keratinocytes. STORM imaging using three independent NUP antibodies showed no statistically significant differences in nuclear pore numbers in differentiation. A mild reduction of NUP93's NPC incorporation was observed in differentiation; however, this NUP93 reduction is not accompanied by drastic differences in nuclear pore permeability or nuclear protein transport, using reporter assays. With NUP93 knockdown in keratinocytes cultured in the progenitor state, transcriptome profiling identified upregulation of keratinocyte differentiation genes related to both keratinization and innate immunity. Consistently, we observed significant nuclear enrichment of NF-κB transcription factors with NUP93 knockdown, together with the upregulation of NF-κB direct target genes and increased NF-κB reporter activity. Taken together, our findings highlight the regulatory roles contributed by the differential NUP expression levels in the progenitor differentiation process.

## Results

**NUP-encoding genes are downregulated in keratinocyte differentiation.** To determine how NPC function is involved in progenitor differentiation, we started by examining the mRNA expression of 30 NUP-encoding genes in primary human keratinocytes in the progenitor state (UD) versus the differentiation state (DF) (4 days in 1.2 mM exogenous $CaCl_2$ with confluency, corresponding to a middle time point in the entire course of six-day keratinocyte differentiation[11,19,22,23]), leveraging the RNA-seq data that we recently generated[11]. The majority (90%) of these NUP genes are downregulated in the differentiation state, including NUP133 that is essential for NPC assembly[24]. NUP93 and NUP205, two inner-ring NUPs, stood out as the most drastically downregulated NUPs in differentiation at the mRNA level (Fig. 1a–c). To validate these findings, we performed qRT-PCR and western blotting for NUP133, NUP93, and NUP205 in the time course of progenitor state (UD), Day 2 (D2) and Day 4 (D4) of keratinocyte differentiation. All these three NUPs showed progressive downregulation at both the mRNA and the protein levels in the differentiation time course (Fig. 1d–i). To determine if this NUP downregulate occurs early in the differentiation process, we compared NUP expression in keratinocytes cultured in low, sub, and full confluence. Confluence is well established as a factor driving keratinocytes' commitment to differentiation, with the induction of the early differentiation marker gene, KRT1 (Keratin 1)[25]. NUP93, NUP205, and NUP133 all showed progressive downregulation with the increase of confluence, at the mRNA level (Supplementary Fig. 1). These data indicate that the transition from the progenitor state towards differentiation for keratinocytes involves NUP downregulation, and that NUP downregulation is an early event in keratinocyte differentiation.

**STORM Imaging identified comparable NPC numbers in differentiation.** NUP downregulation in keratinocyte differentiation suggested a possibility of NPC number reduction, although NUP proteins also participate in other biological processes other than NPC assembly[26]. To test this, we applied Stochastic Optical Reconstruction Microscopy (STORM), with spatial resolution at ~25 nm, to resolve individual NPCs and enable quantification of endogenous NUP proteins. As STORM imaging relies on high-quality antibody binding, we implemented the Nuclei-Isolation Staining (NIS) method[27] to enhance antibody nuclear accessibility. NIS reduces STORM imaging background and enhances imaging quality, by removing cytoplasmic proteins that can contribute to non-specific antibody binding. Furthermore, NIS bypasses the highly crosslinked cytoplasmic proteins (associated with the barrier function) in the differentiated keratinocytes, equalizing the antibody-labeling efficiency of NUP proteins on the nuclear envelope between the progenitor state and the differentiation state. To acquire STORM images under the same experimental conditions for these two states, we used a silicone spacer with multiple wells on top of a poly-lysine treated coverslip

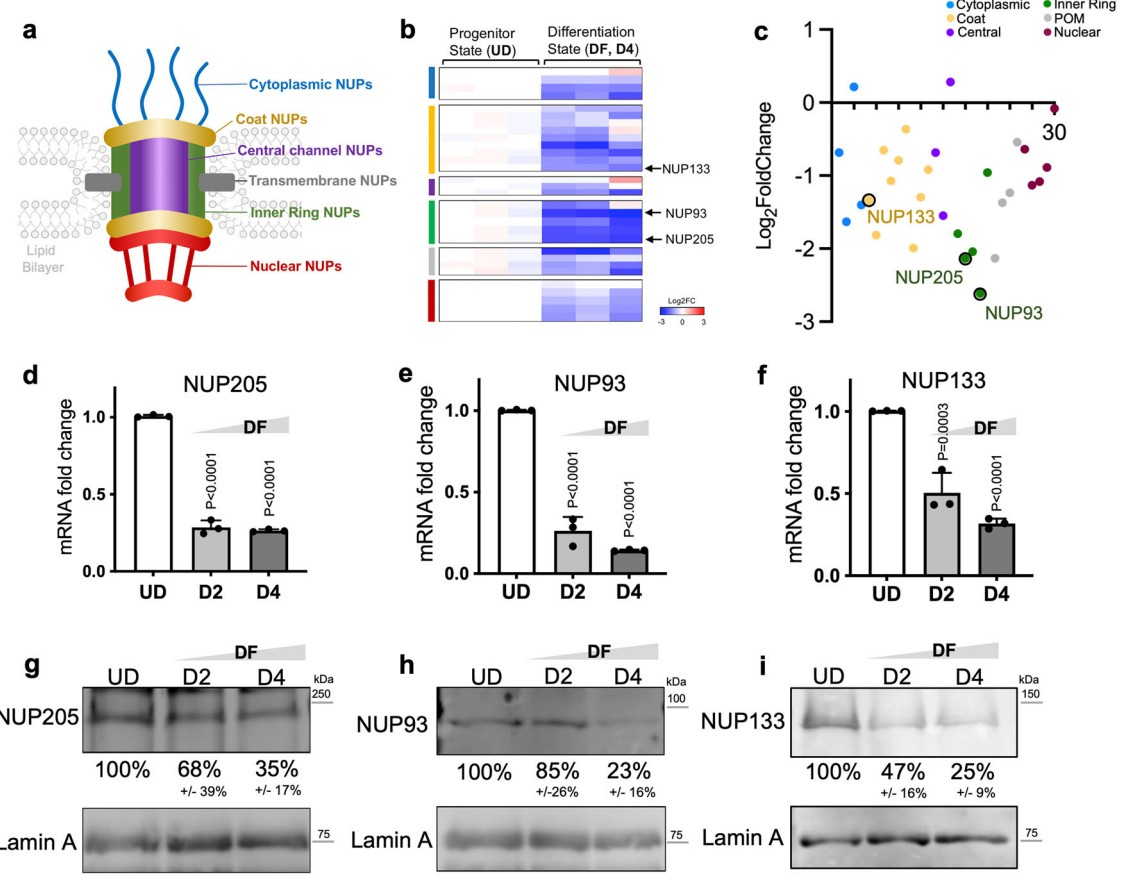

**Fig. 1 Downregulation of NUP-encoding genes in keratinocyte differentiation. a** Graphical representation of the nuclear pore complex (NPC) embedded in the lipid bilayers (light gray). The 6 different subcategories of NUPs are color coded and labeled. **b** Heatmap of RNA-seq data comparing NUP expression in undifferentiated (UD) and differentiated (DF) keratinocytes in triplicates, separated in the 6 categories using the color coding scheme as in the graphic representation. **c** Visualization of the relative changes of NUP expression in keratinocyte differentiation, with each dot representing a NUP gene color-coded according to the NUP location in the graphic representation. **d–f** RT-qPCR of NUP205, NUP93 and NUP133 mRNA expression in undifferentiated (UD), day 2 of differentiation (D2), and day 4 of differentiation (D4). (one-way ANOVA with post-hoc test, $N = 3$ biological replicates, data are represented as mean ± standard deviation.) **g–i** Western blotting comparing NUP205, NUP93 and NUP133 protein levels in differentiation time course, with Lamin A as a loading control. Quantifications of NUP protein expression levels are presented as average $+/-$ SD, $n = 3$.

(Fig. 2a). Nuclei were extracted simultaneously from keratinocytes in the progenitor state and the differentiation state, immobilized in different wells on the same coverslip. Staining and imaging procedures were performed with the two set of nuclei side-by-side. During the imaging process, the top of the silicon spacer was sealed with another piece of glass coverslip, preventing oxygen exchange from the air to minimize dye photobleaching during STORM imaging.

To determine if keratinocyte differentiation involves reduced NPC numbers on the nuclear envelope, we used the MAB414 antibody that recognizes multiple NUPs featuring the FXFG repeats[28], including NUP62, NUP214, NUP358, and NUP153 (Fig. 2b). STORM images generated using MAB414 showed distinct dot-like clusters on the surface of the nuclei, consistent with the detection of coat, inner-ring, and nuclear-basket NUPs by this antibody, in contrast to the blurry image acquired using conventional epifluorescence microscopy (Fig. 2c, d). To determine NPC numbers based on this MAB414 STORM imaging, we adapted an unsupervised learning-based clustering analysis method (Hierarchical Density-Based Spatial Clustering of Applications with Noise, or HDBSCAN) to extract individual NPC[29]. The minimal number of points per cluster for HDBSCAN was optimized using Monte Carlo simulation. More specifically, the coordinates of single-molecule localizations (SML), which

represent the photo-switching events that arose from single fluorescent molecules bound to the individual NPCs, were assigned to separate clusters (as the blue-, green- and red-colored dot clusters in Fig. 2e). From differentiated keratinocytes, we observed very similar dot-like features using the MAB414 antibody on the surface of isolated nuclei (Fig. 2f, g). To compare NPC numbers represented by MAb414 in the progenitor state versus the differentiation state, we first quantified the density of NPC cluster numbers extracted using HDBSCAN (Fig. 2h), and the cluster density was not statistically significant between the two states. We also compared the sizes of the nuclei extracted from the two states of keratinocytes and did not observe a significant difference (Supplementary Fig. 2a). These data suggest that the NPC number is not significantly between the two states, detected by MAB414. In addition, we analyzed the STORM data to compare density of SML in each image, the number of SML assigned to each NPC cluster, and the size of the NPC clusters labeled by MAB414 (Fig. 2i–k). No significant differences were identified in all these comparisons.

Given the characteristic ring structure of NPC representable by NUP133 established super-resolution imaging in immortalized cell lines[30], we used an NUP133 antibody to label the endogenous protein in the nuclei extracted from the keratinocytes, as an orthogonal approach to MAB414. NUP133 is recognized as part

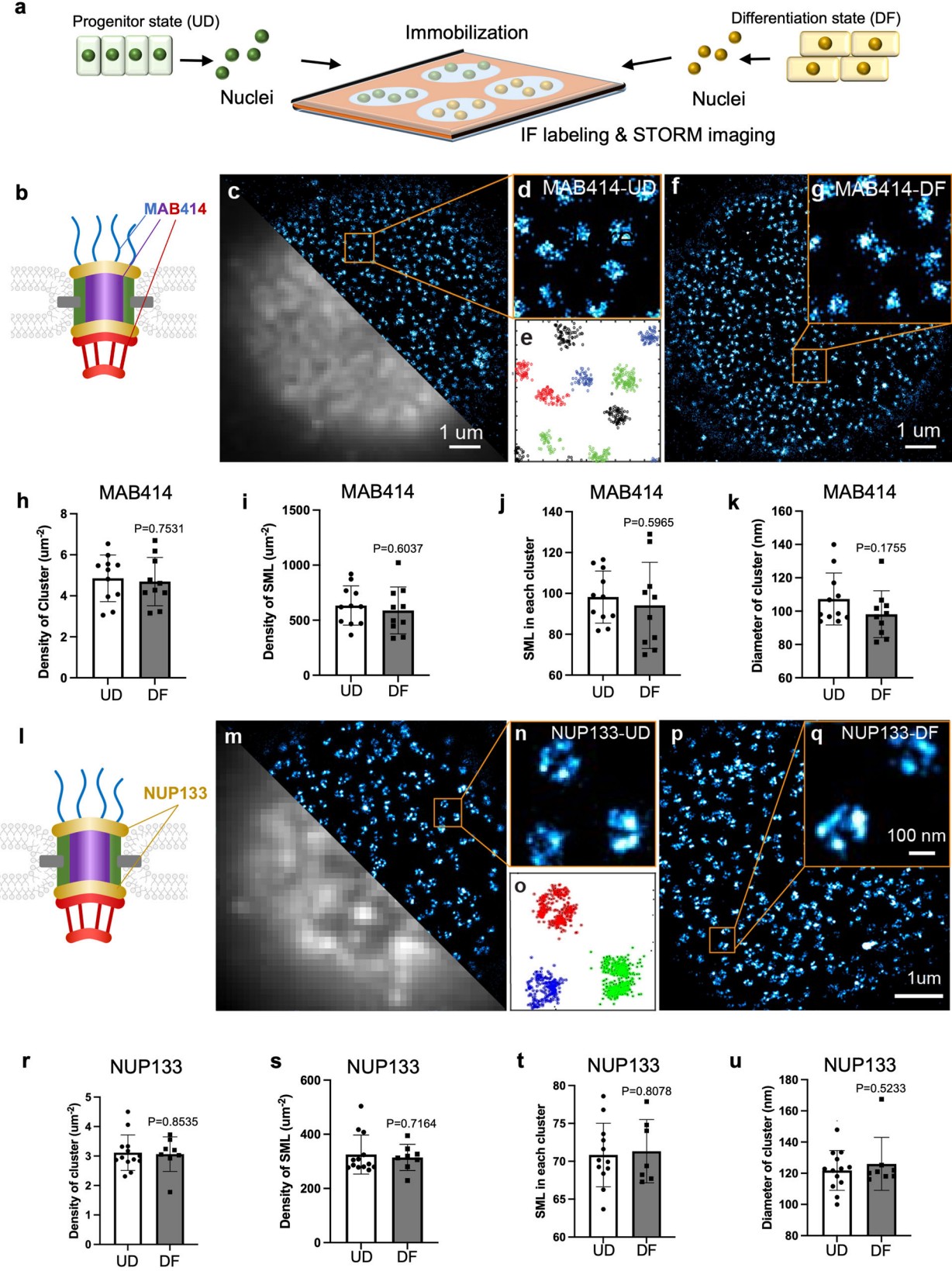

of the Y-complex, which is part of the coat NUPs that localize to both the cytoplasmic and nuclear sides of the nuclear pore[6,7,24,31] (Fig. 2l). Our STORM imaging fully resolved the ring-like feature of individual NPCs at the surface of the nuclei, in contrast to the blurred images acquired (Fig. 2m, n). Using HDBSCAN, we quantified the NUP133 clusters in the progenitor state versus the differentiation state, and identified no statistically significant differences (Fig. 2o–r). Quantification of SML density, SML in each NPC cluster, and NPC cluster sizes again showed no significant differences between the progenitor and differentiated states (Fig. 2s–u). The sizes of the nuclei labeled by NUP133 also showed no significant difference (Supplementary Fig. 2b). Thus,

**Fig. 2 Comparable NPC numbers in differentiation identified by STORM imaging. a** Illustration of the methodology designed to perform STORM imaging of isolated nuclei, comparing the progenitor-state versus the differentiate-state keratinocytes. **b** Illustration showing the relative location of the NUPs recognized by MAB414 in an NPC. **c** Representative images of MAB414 antibody labeling for the progenitor-state keratinocytes, comparing the imaging quality between STORM (top) and epifluorescence (bottom). **d** Magnified view of the boxed region in **c**, showing cluster structures labeled by the MAB414 antibody. **e** A scatterplot showing HDBSCAN clustering results representing the distinct NPCs colored in black, red, green, and blue. **f** A representative STORM image of MAB414 antibody labeling for the nucleus from differentiation-state keratinocyte. **g** Magnified view of the boxed region in **f**. **h** Quantification of NPC density comparing MAB414 in UD ($n = 11$) and DF ($n = 9$) calculated by HDBSCAN. $p = 0.75$, $t$-test. Data are represented as mean ± standard deviation. **i** Quantification of the overall spatial density of SML of MAB414 staining in UD and DF. $p = 0.60$, $t$-test. Data are represented as mean ± standard deviation. **j** Quantification of SML within each NPC cluster in UD and DF nuclei. $p = 0.60$, $t$-test. Data are represented as mean ± standard deviation. **k** Quantification of the diameter of the MAB414 clusters in UD and DF. $p = 0.18$, $t$-test. Data are represented as mean ± standard deviation. **l** Graphical representation of the localization of NUP133, part of the Y complex, in NPC. **m** A representative STORM image of a UD nucleus stained with NUP133. **n** Magnified view of the boxed region in **m**, showing ring-like structures labeled by the NUP133 antibody. **o** A scatterplot showing HDBSCAN clustering results representing the three distinct NPCs colored in red, green, and blue. **p** A representative STORM image of a DF nucleus labeled with MAB414. **q** Magnified view of the boxed region in **p** showing NPC ring structures. **r** Quantification of NPC density comparing NUP133 in UD and DF clustered by HDBSCAN under identical input parameters. $p = 0.85$, $t$-test. Data are represented as mean ± standard deviation. **s** Quantification of the overall spatial density of SML of NUP133 staining in UD ($n = 13$) and DF ($n = 8$). $p = 0.72$, $t$-test. Data are represented as mean ± standard deviation. **t** Quantification of SML within each NPC cluster in UD and DF nuclei. $p = 0.81$, $t$-test. Data are represented as mean ± standard deviation. **u** Quantification of the diameter of the NUP133 clusters in UD and DF. $p = 0.5$, $t$-test. Data are represented as mean ± standard deviation.

NPC quantification using two different antibodies consistently showed no statistically significant changes in NPC numbers between the progenitor-state and differentiation-state keratinocytes.

**NUP93 reduction in differentiation is independent of NPC numbers**. NUP93 is the most downregulated NUP at the mRNA level in keratinocyte differentiation, with the downregulation further validated at the protein level. To determine if NUP93's incorporation to NPC's is reduced in keratinocyte differentiation, we performed super-resolution imaging using an NUP93 antibody in the progenitor state versus the differentiation state. STORM images of NUP93 showed dot-like clusters on the surface of the nuclei (Fig. 3a–e), similar to the clusters labeled by MAB414. The cluster number and diameter did not significantly change in differentiation (Fig. 3f, g), consistent with the findings from NUP133 or MAB414 labeling, suggesting that the NPCs labeled by NUP93 did not significantly change their numbers in differentiation. However, SML density and the SML in each NPC cluster were significantly reduced in the differentiation state (Fig. 3h, i), suggesting that NUP93 incorporation into individual NPCs is decreased in keratinocyte differentiation.

Although NUP93's incorporation to NPC is statistically significantly reduced in differentiation, an average of 72% of NUP93 is still estimated to retain NPC incorporation in the differentiation state. However, our western blotting data using whole cells estimate only 23% of NUP93 protein level is retained in the differentiation state. Seeking an explanation for the differences in the relative NUP93 protein level in differentiation at the nuclear pores versus the whole cells, we coupled confocal imaging with subcellular fractionation. Confocal imaging in progenitor-state keratinocytes identified that NUP93 also localizes in the cytoplasm and the nucleoplasm (Supplementary Fig. 2c). To compare the progenitor state with the differentiation state, we fractionated keratinocytes in these two states into the nuclear and cytoplasmic fractions and performed western blotting. Interestingly, NUP93 showed significantly higher enrichment in the cytoplasmic fraction in the progenitor state than the differentiation state (Fig. 3j, k). The fraction of NUP93 in the extracted nuclei also showed about 4 times higher enrichment in the progenitor state than the differentiation state (Fig. 3l). These data suggest that the NUP93 fractions not associated with the nuclear pores are more drastically reduced in keratinocyte differentiation.

NUP133 is also significantly reduced at the protein level in the whole-cell western, but did not show drastic changes at the nuclear pore in our STORM analysis. To clarify this, we also applied both confocal microscopy and subcellular fractionation. Confocal imaging of NUP133 identified strong signals in the cytoplasm, in addition to the nuclear envelope (Supplementary Fig. 2d). Similar to NUP93, NUP133 showed significantly higher enrichment in the cytoplasmic fraction in the progenitor state than the differentiation state (Fig. 3m, n). However, the NUP133 nuclear fraction is not significantly enriched in the progenitor state (Fig. 3o), consistent with the confocal imaging data that NUP133 is less enriched in the nucleoplasm as compared to NUP93. Thus, NUP133 reduction in the cytoplasmic fraction contributes to its overall reduction in keratinocyte differentiation.

**NUP93 knockdown impaired progenitor maintenance**. To investigate how NUP93 reduction influences progenitor differentiation, we used RNAi to knockdown NUP93 in the progenitor-state keratinocytes. Four independent shRNAs targeting NUP93 were expressed individually in keratinocytes cultured in the progenitor-state condition. All four shRNAs effectively reduced NUP93 expression at the mRNA and protein levels (Fig. 4a–c). To determine if the high levels of NUP93 is essential for progenitor maintenance, we first performed a clonogenicity assay[32]. Equal numbers of keratinocytes with NUP93 knockdown or control were seeded onto a layer of 3T3 feeder cells, and the number of visible clones that originated from single cells were quantified after 2 weeks. NUP93 knockdown drastically reduced the clone numbers (Fig. 4d, e), indicating impaired keratinocyte clonogenicity. We subsequently performed progenitor competition assay to evaluate the capacity of keratinocytes to regenerate three-dimensional epidermal tissue. GFP-labeled and DsRed-labeled keratinocytes were mixed at 50:50 and seeded onto a piece of devitalized human dermis raised at the air-liquid interface. In all the experimental groups, the GFP-labeled keratinocytes co-expressed control shRNA. The DsRed-labeled keratinocytes co-expressed either control shRNA or NUP93 shRNA. After 6 days of regeneration, the tissues were sectioned and the number of red or green keratinocytes in the basal progenitor layer were quantified. The expression of NUP93-shRNA, but not control shRNA, strongly reduced the representation of Ds-Red labeled keratinocytes in the basal progenitor layer of the epidermal tissue (Fig. 4f, g). These data indicate that the high NUP93 level is essential for sustaining the regenerative capacity of the progenitor-state keratinocytes.

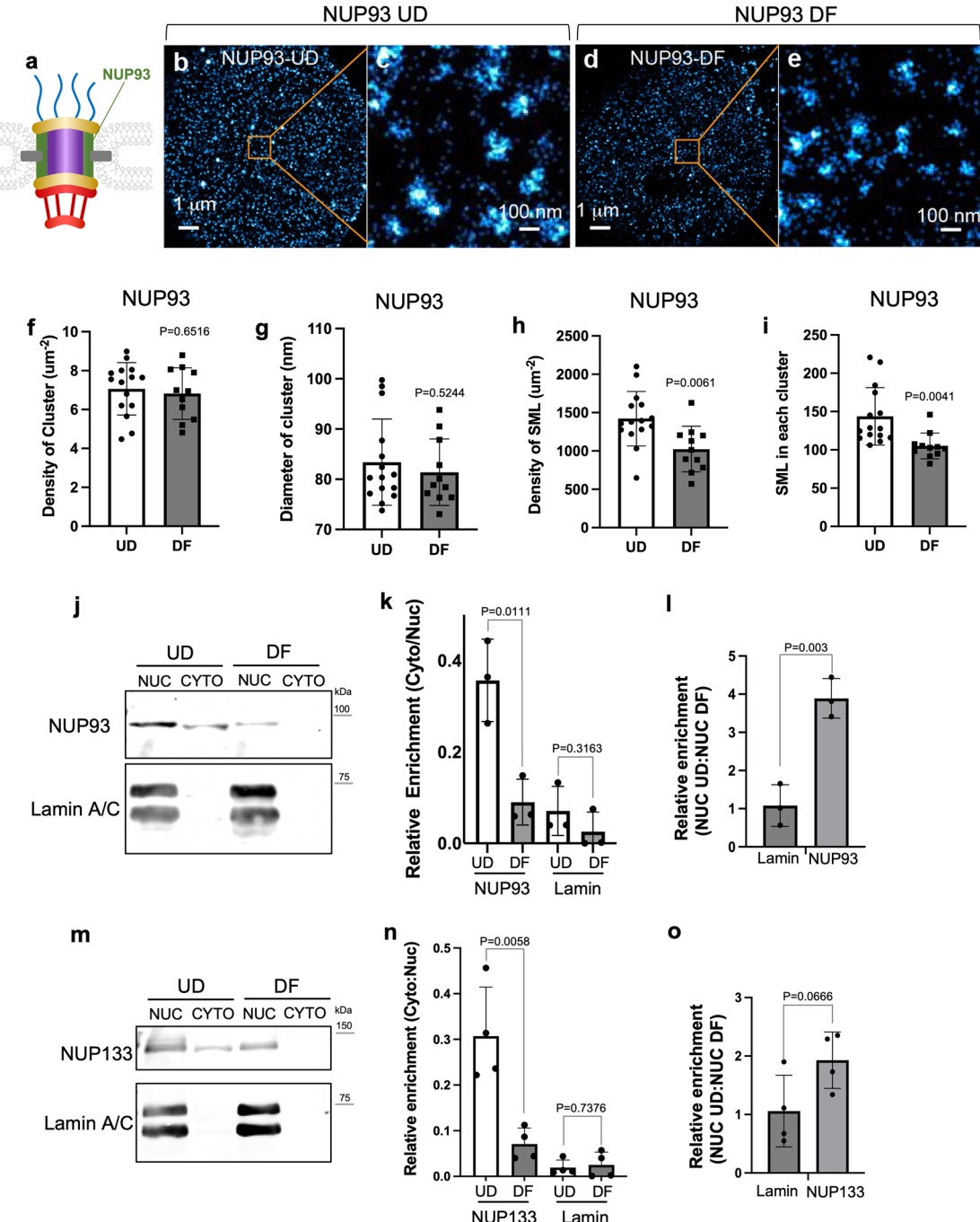

**NUP93 reduction does not drastically alter NPC numbers or basic transport function**. To investigate how NUP93 reduction influences keratinocyte function, we first examined the number of NPCs in keratinocytes with NUP93 knockdown versus control using MAB414. Consistent with our findings comparing the progenitor state versus the differentiation state, NUP93 knockdown did not lead to significant changes in NPC numbers (Fig. 5a–e).

As the inner-ring NUP93 was implicated in establishing the nuclear pore passive permeability[33,34], we asked if the passive transport is altered with NUP93 reduction in primary human keratinocytes with knockdown or in the differentiation process. To test this, we designed a Fluorescence Recovery After Photobleaching (FRAP) experiment. We expressed both mCherry (28 KDa, freely diffusing between the nucleus and the cytoplasm through

the nuclear pores) and H2B-GFP (marking the nucleus) in primary human keratinocytes. The mCherry fluorescence in the nucleus was laser bleached with a confocal microscope and the time for fluorescence to recover in the nucleus was recorded and quantified. The half time for fluorescence recovery in the nucleus in keratinocytes with NUP93 knockdown was not significantly different from the non-targeting control (Fig. 5f, g). Similarly, the difference of FRAP half time in the differentiation state versus the progenitor state was not significantly different (Fig. 5h, i). These data suggest that NUP93 reduction, either through knockdown or in differentiation, does not drastically alter the passive permeability of the nuclear pores.

To determine if NUP93 reduction in keratinocytes affects the nuclear import function associated with the nuclear pores, we constructed an NLS-LacZ-mCherry reporter (Supplementary

**Fig. 3 Reduced NUP93 incorporation into NPCs in differentiation. a** Illustration showing the relative location of NUP93 in the NPC. **b** Representative STORM image of NUP93 labeling on the surface of a nucleus from UD keratinocytes. **c** Magnified view of the boxed region in **b** showing NUP93 clusters. **d** Representative STORM image of NUP93 labeling on the surface of a nucleus from DF keratinocytes. **e** Magnified view of the boxed region in **d** showing NUP93 clusters. **f** Quantification of NPC density comparing NUP93 in UD and DF calculated by HDBSCAN. $p = 0.6516$, t-test. Data are represented as mean ± standard deviation. **g** Quantification of the diameter of the NUP93 clusters in UD and DF. $p = 0.5244$, t-test. Data are represented as mean ± standard deviation. **h** Quantification of the overall spatial density of SML of NUP93 staining in UD ($n = 15$) and DF ($n = 11$). $p = 0.0061$, t-test. Data are represented as mean ± standard deviation. **i** Quantification of SML within each NPC cluster in UD and DF nuclei. $p = 0.0041$, t-test. Data are represented as mean ± standard deviation. **j** Representative western-blot image showing NUP93 protein levels in the nuclear and cytoplasmic fractions of keratinocytes, comparing the progenitor state versus the differentiation state. Lamin A/C was used as a loading control for the nuclear fraction. **k** Quantification of the ratio of NUP93 protein levels detected in the cytoplasmic fraction vs the nuclear fraction (Cyto:Nuc). $n = 3$. The Cyto:Nuc distribution of lamin A/C shows no significant difference between the UD and DF states ($p = 0.3163$, t-test). The Cyto:Nuc distribution of NUP93 is significantly higher in the UD state than the DF state ($p = 0.0111$, test). Data are represented as mean ± standard deviation. **l** Quantification of the ratio of nuclear NUP93 in UD versus DF state. There is a significant reduction of nuclear NUP93 in the DF state ($p = 0.003$, t-test). Data are represented as mean ± standard deviation. **m** Representative western-blot image showing NUP133 protein levels in the nuclear and cytoplasmic fractions of keratinocytes, comparing the progenitor state versus the differentiation state. Lamin A/C was used as a loading control for the nuclear fraction. **n** Quantification of the ratio of protein levels detected in the cytoplasmic fraction vs the nuclear fraction (Cyto:Nuc). $n = 4$. The Cyto:Nuc distribution of lamin A/C shows no significant difference between the UD and DF states ($p = 0.74$, t-test). The Cyto:Nuc distribution of NUP133 is significantly higher in the UD state than the DF state ($p = 0.016$, t-test). Data are represented as mean ± standard deviation. **o** Quantification of the ratio of nuclear NUP133 in UD versus DF state. There is not a significant reduction of nuclear NUP133 in the DF state ($p = 0.0666$, t-test). Data are represented as mean ± standard deviation.

Fig. 3a). As expected, this fusion protein localized predominantly inside the nucleus. After the fluorescence within the nucleus was photobleached, the fluorescence gradually recovers as the cytoplasmic portion was actively imported to the nucleus. With NUP93 knockdown, the 2 shRNAs did not consistently show significant changes in recovery kinetics quantified by the half time of recovery (Supplementary Fig. 3b, c). Between the progenitor state and the differentiation state, no statistically significant difference was observed (Supplementary Fig. 3d, e). These data suggest that NUP93 reduction in keratinocyte is not associated with drastically altered kinetics with nuclear import.

In addition to the nuclear import function, we investigated if NUP93 reduction affects nuclear export. We constructed an NES-mCherry reporter and expressed it in keratinocytes (Supplementary Fig. 4a). This reporter protein predominantly localized in the cytoplasm of keratinocytes, as expected. Treating keratinocytes with LMB, which inhibits the major export protein CRM1, led to the translocation of this NES-mCherry reporter into the nucleus (Supplementary Fig. 4b). NUP93 knockdown in keratinocytes did not drastically change the subcellular localization of this NES-mCherry reporter (Supplementary Fig. 4c), suggesting NUP93 knockdown did not impair the general function of nuclear export. Apoptosis assay using Mitoview indicate that NUP93 knockdown does not led to rapid induction of massive apoptosis in keratinocytes (Supplementary Fig. 4d).

**NUP93 knockdown de-repressed keratinocyte differentiation gene expression**. To determine how NUP93 knockdown impairs progenitor maintenance, we performed transcriptome profiling using RNA-seq. In total, we identified 2118 differentially expressed genes with NUP93 knockdown (fold change >2, $p < 0.05$, Fig. 6a, Supplementary Data 1). 62% of these genes were upregulated, with the top than downregulated genes (38%). Interestingly, the top KEGG pathway associated with these upregulated genes was TNF signaling pathway, while the pathways associated with the downregulated genes are related to metabolic pathways and cell cycle (Fig. 6b). In addition, the expression of other NUPs were not drastically impacted by NUP93 knockdown (Supplementary Fig. 5a). For example, NUP205 and NUP188 were previously shown to be part of the NUP93 complex, and both are strongly downregulated in differentiation (average Log2 fold change −2.2 and −1.8, respectively). In NUP93 knockdown, these two NUPs did not pass our fold change cutoff (average Log2 fold change −0.5 and −0.8,

respectively), suggesting that NUP93 reduction by itself is not sufficient to explain the drastic NUP reduction in the differentiation process.

To determine if NUP93 knockdown could account for a subset of gene expression changes in the keratinocyte differentiation process that features NUP93 reduction, we compared these 2118 genes with the calcium-induced differentiation signature[11]. A total of 1180 genes are shared between these two data sets ($p < 0.0001$, Fisher's exact test). The top GO terms of the shared upregulated genes are associated with keratinization and innate immunity. The top GO terms for the downregulated genes are related to cell cycle (Fig. 6c, d). Using qRT-PCR, we confirmed the induction of representative genes related to keratinocyte differentiation (*OVOL1, S100A9*) and immune response genes (*NOD2, IL36G*) with NUP93 knockdown or in differentiated keratinocytes (Fig. 6e–h). These data suggest that NUP93 reduction contribute to the induction of the mechanical and immune barrier functions that occur in keratinocyte differentiation.

Since NUP93 knockdown upregulated genes related to the differentiation process, we examined if NUP93 overexpression may block the differentiation process. We introduced an NUP93 overexpression construct in keratinocytes, which resulted in around 6-fold upregulation of NUP93 protein level. These keratinocytes were then subjected to the differentiation culture condition (confluence and 1.2 mM $CaCl_2$ for 4 days). Using qRT-PCR, we compared the expression of representative differentiation genes, but did not detect drastic changes (Supplementary Fig. 5b–d). These findings suggest that the high level of NUP93 is necessary but not sufficient to repress many differentiation genes in the process of keratinocyte differentiation.

A key pathway activating both the mechanical and immune barrier function is NF-κB. As the upregulated genes in NUP93 knockdown are related to the TNF signaling and innate immunity, we further examined the expression of multiple known NF-κB target genes. We found that these are upregulated in both NUP93 knockdown and in the differentiation process (Fig. 6i). These data suggest that the NF-κB pathway is activated with NUP93 knockdown in keratinocytes cultured in the progenitor condition. In addition, we compared the RNA-seq data of NUP93 knockdown with the knockdown of two other NUPs, NUP98 and RAE1[3]. While the knockdowns of these NUPs share a signature of upregulating differentiation genes and downregulating cell cycle genes, NUP93 knockdown uniquely upregulates genes related to innate immunity (Supplementary

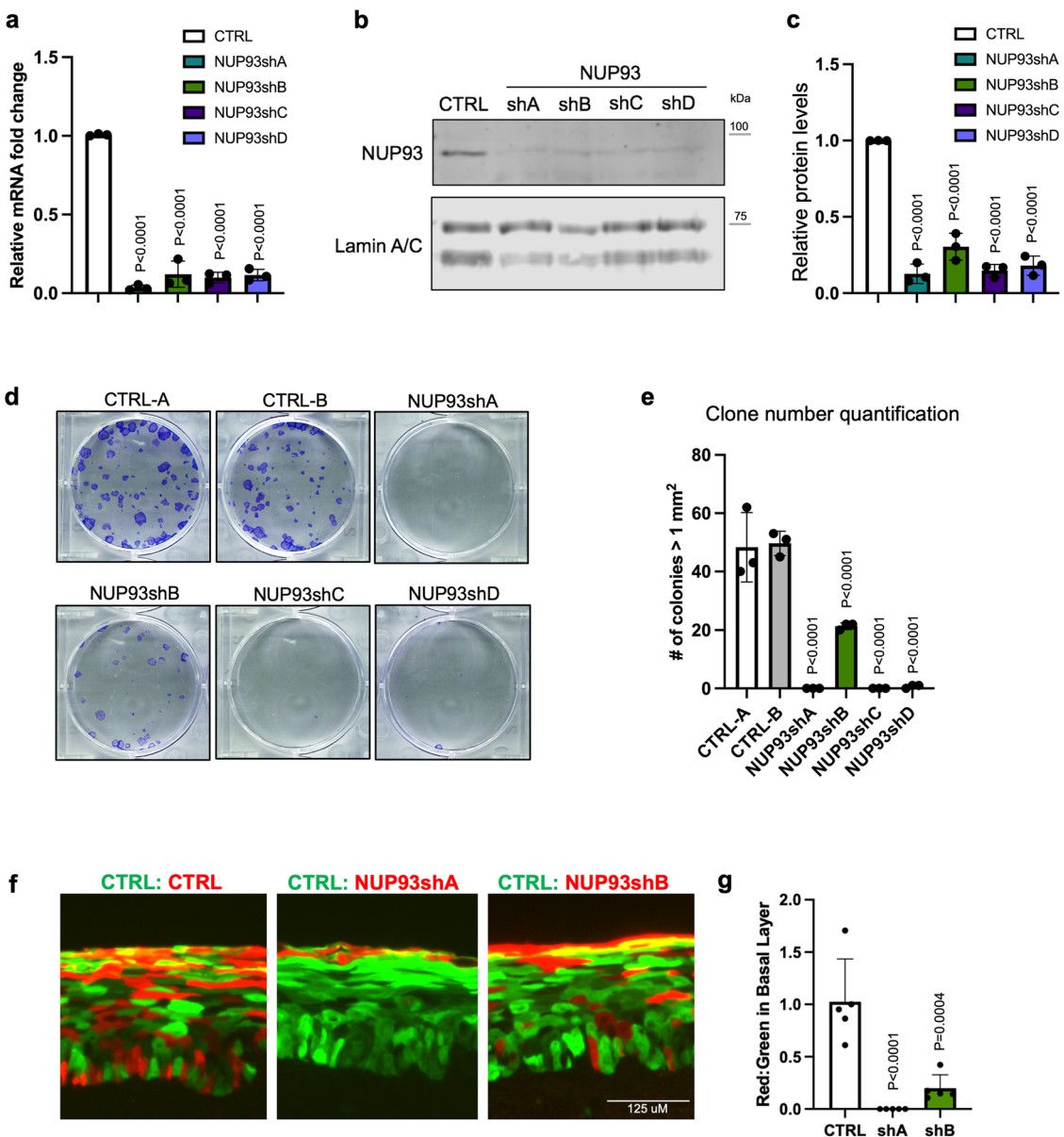

**Fig. 4 NUP93 knockdown diminishes progenitor regenerative capacity. a** qRT-PCR quantification of NUP93 knockdown efficiency by shA, shB, shC, or shD versus control. (One-way ANOVA with post-hoc test, $N = 3$ biological replicates.) Data are represented as mean ± standard deviation. **b** Western blots showing the knockdown efficiency of NUP93 shRNAs at the protein level. **c** Quantitation of NUP93 protein levels, relative to the lamin A/C loading control, is presented as average $+/-$SD, $n = 3$. **d**, **e** Representative images and quantification of keratinocyte clonogenicity, comparing NUP93 knockdown versus control average (one-way ANOVA with post-hoc test, $N = 3$ biological replicates.) Data are represented as mean ± standard deviation. **f** Representative images of regenerated epidermal issue initiated with a 50:50 mix of DsRed and GFP labeled keratinocytes. The GFP-labeled keratinocytes co-expressed a non-targeting control shRNA. The DsRed-labeled keratinocytes coexpressed control shRNA, NUP93 shA, or NUP shB. **g** The ratio of DsRed-labeled keratinocytes versus GFP-labeled keratinocytes in the basal progenitor layer of the regenerated epidermal tissue. This ratio was drastically reduced when DsRed-labeled keratinocytes co-expressed NUP93 shA or NUP93 shB, but not when the DsRed-labeled keratinocytes co-expressed non-targeting control shRNA. Data are represented as mean ± standard deviation.

Fig. 5e–j). These data suggest that the regulation of immune related genes is uniquely modulated by NUP93, but not all NUPs.

**NUP93 knockdown in the progenitor state enhanced NF-κB nuclear translocation.** NF-κB activation depends on the nuclear translocation of transcription factors such as p65 and p50, and increased nuclear localization of the NF-κB transcription factor p50 had been reported in epidermal differentiation[35]. To determine if NUP93 knockdown alters the subcellular localization of NF-κB transcription factors, we performed immunofluorescence staining. In normal progenitor-state keratinocytes, p65 is enriched in the

cytoplasm. Strikingly, keratinocytes with NUP93 knockdown exhibited drastically increased nuclear translocation of p65 as compared with the non-targeting control (Fig. 7a, b). Similarly, p50 also showed an increase of the N:C ratio with NUP93 knockdown (Fig. 7c, d). This change of subcellular NF-κB was not a result of differences in keratinocyte confluency between NUP93 knockdown versus control, as normal keratinocytes grown in low versus high level confluency consistently exhibit predominant cytoplasmic localization of p65 and p50 (Supplementary Fig. 6a). Additionally, NUP93 overexpression in keratinocytes did not result in drastic changes of p65/p50 subcellular localization (Supplementary Fig. 6b).

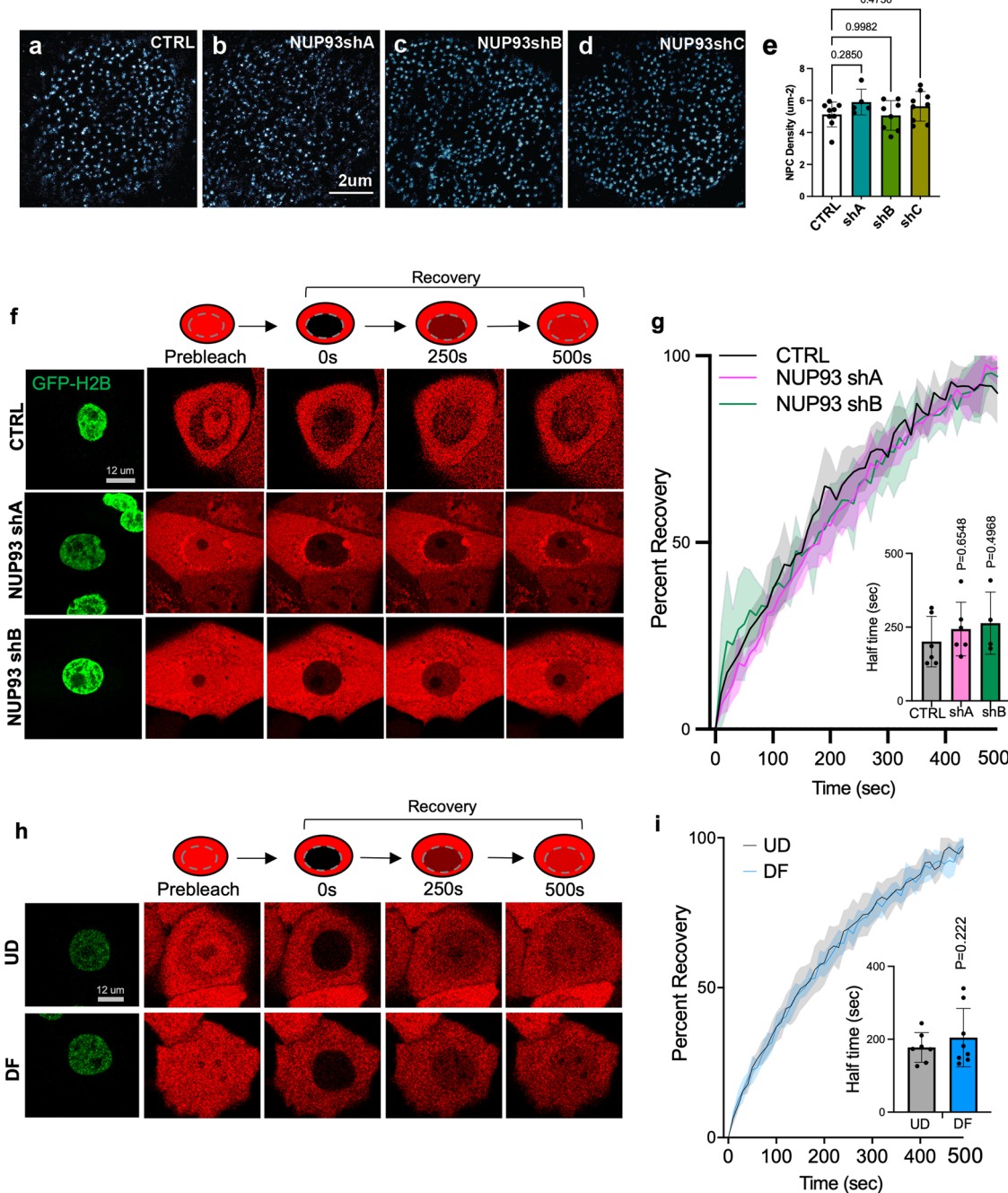

**Fig. 5 NUP93 reduction does not significantly affect nuclear pore number or permeability. a–d** Control knockdown versus NUP93 knockdown of MAB414 STORM imaging. **e** Quantification of NPC density in control and NUP93 knockdown with MAB414 STORM imaging. ($n > 5$, $t$-test.) Data are represented as mean ± standard deviation. **f** Graphical illustration of the FRAP method for measuring nucleocytoplasmic transport dynamics, and representative cell images from the FRAP experiment comparing keratinocytes with NUP93 knockdown versus control. **g** Fluorescence recovery trajectory of mCherry in the nucleus after photobleaching, comparing keratinocytes with or without NUP93 shA or shB ($n = 6$/condition). Line and shaded area represent mean and standard errors. Quantification of half time for recovery is included as a bar graph (one-way ANOVA with post-hoc test, and data are represented as mean ± standard deviation). **h** Representative cell images from the FRAP experiment comparing UD versus DF keratinocytes. **i** Fluorescence recovery trajectory of mCherry in the nucleus after photobleaching, comparing UD ($n = 9$) vs DF ($n = 8$) keratinocytes. Line and shaded area represent mean and standard errors. Quantification of half time for recovery is included as a bar graph (one-way ANOVA with post-hoc test, and data are represented as mean ± standard deviation).

To further determine if this increased nuclear p65 and p50 with NUP93 knockdown correspond to increased NF-κB DNA binding and activity, we leveraged a luciferase reporter system that includes 4 copies of NF-κB consensus binding sites[36]. This reporter was lentivirally delivered to keratinocytes. Keratinocytes with NUP93 knockdown consistently showed increased luciferase activity as compared to the non-targeting control (Fig. 7e), suggesting increased NF-κB activity concomitant with the increased p65/p50 nuclear localization.

We subsequently examined if NUP93 knockdown influences the subcellular localization of other transcription regulators. We examined IKKα (another NF-κB regulator that can localize in the

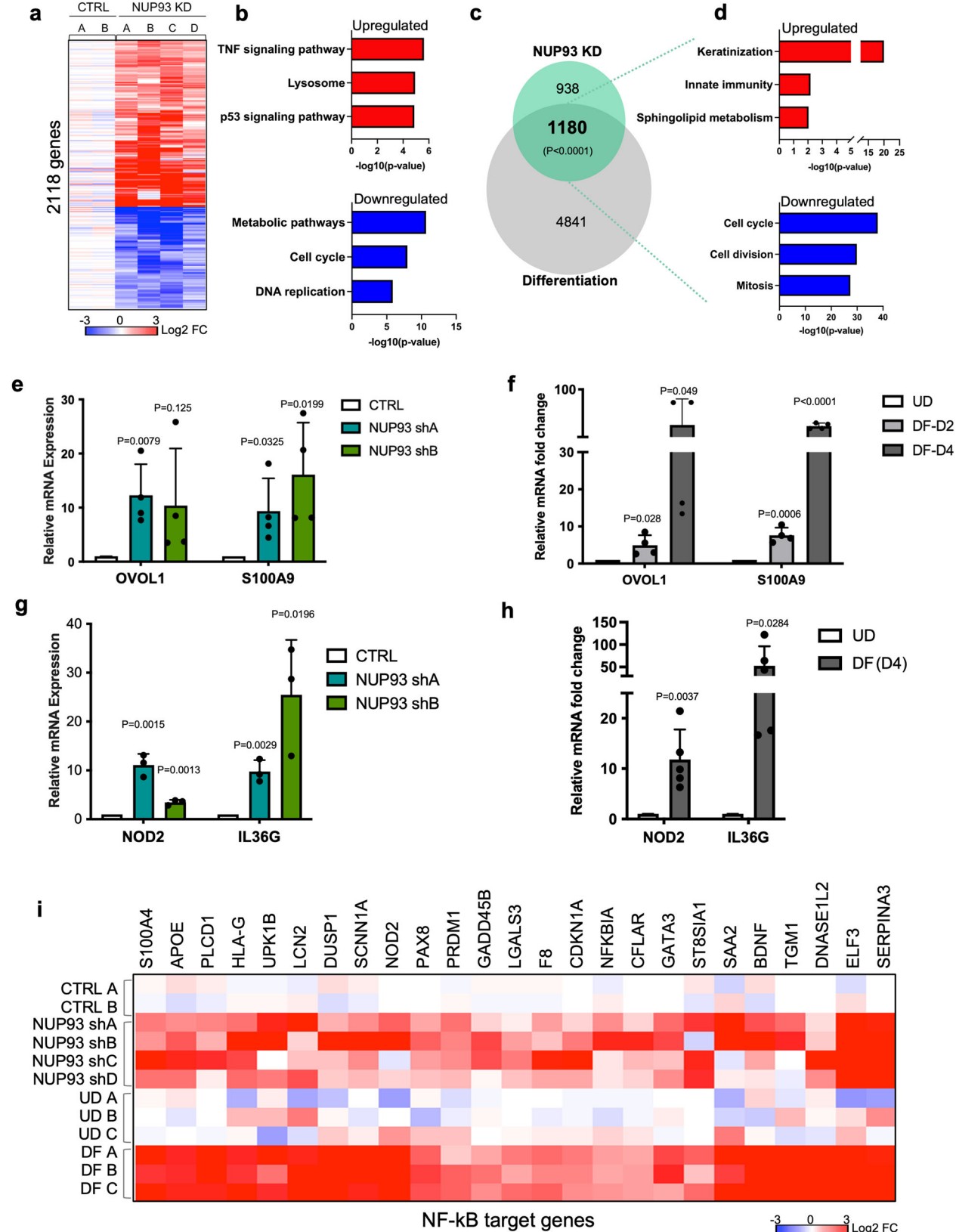

nucleus) and SMAD (another nucleocytoplasmic shutting transcription regulator in the TGF-ß pathway, involved in keratinocyte differentiation) and found no drastic changes of N:C ratio, with only a subtle increase with shA and no significant change with shB (Supplementary Fig. 6c–f). These data indicate that the nuclear localization of the NF-κB transcription factors

p65 and p50, but not all nucleocytoplasmic-shutting transcription regulators, were drastically influenced by NUP93 knockdown.

This unique nuclear enrichment of p65 and p50 with NUP93 knockdown, but not other transcription regulators we examined, is consistent with the reporter-assay results that the general nuclear pore permeability and import/export function was not

**Fig. 6 NUP93 knockdown in the progenitor state de-represses differentiation programs. a** Heatmap showing the relative expression of 2118 differentially expressed genes in keratinocytes with NUP93 knockdown versus control, cultured in the progenitor condition. **b** Top Gene Ontology (GO) terms associated with these significantly changed genes with NUP93 knockdown. **c** Overlap of NUP93 differentially expressed genes and genes differentially expressed if keratinocyte differentiation (Fisher's exact test). **d** Top GO terms associated with the shared upregulated or downregulated overlapping gene**s** between NUP93 knockdown and differentiation. **e, f** qRT-PCR quantification of representative differentiation marker genes in keratinocytes, comparing NUP93 knockdown versus control ($n = 4$), or UD versus DF keratinocytes ($n = 4$). (One-way ANOVA with post-hoc test, $N = 4$ biological replicates). Data are represented as mean ± standard deviation. **g, h** qRT-PCR quantification of representative genes with immune-barrier functions, comparing NUP93 knockdown versus control ($n = 3$), or UD versus DF keratinocytes ($n = 5$). (one-way ANOVA with post-hoc test) Data are represented as mean ± standard deviation. **i** Heatmap showing the relative expression of NF-κB target genes in keratinocytes with NUP93 knockdown or in differentiation, based on the RNA-seq data.

altered. To investigate the potential mechanisms underlying this unique impact on p65/p50, we re-examined the images of keratinocytes expressing the mCherry or mCherry-LacZ-NLS reporter. We noticed accumulations of mCherry signals right outside the nucleus in keratinocytes with NUP93 knockdown, but with the non-targeting control (Supplementary Fig. 7a, b), suggesting proteins accumulation in the endoplasmic reticulum (ER). Protein accumulation in the ER was previously linked to NF-κB activation in 293 cells[37]. To determine if protein accumulation in ER could lead to NF-κB activation in primary human keratinocytes, we treated keratinocytes with Brefeldin A, which inhibits protein transport from the ER and leads to protein accumulation[37,38]. We found that the nuclear localization of p50 or p65 was significantly increased upon Brefeldin A treatment in keratinocytes, coupled with increased expression of representative NF-κB target genes (Supplementary Fig. 7c–f). We further identified the GO term of response to unfolded protein ($p = 0.04$) associated with the upregulated genes with NUP93 knockdown. These genes include HSPA6, HSPA7, ERP27, DNAJB2 and CREBRF, which are known to be induced by ER stress[39–43]. Notably, these genes are induced in progenitor-state keratinocytes not only with NUP93 knockdown, but also in the process of differentiation (Supplementary Fig. 7g), in agreement with previous findings that ER stress is involved in keratinocyte differentiation. These findings suggests that ER stress could contribute to the specific induction of NF-κB in progenitor-state keratinocytes upon NUP93 knockdown.

## Discussion

Our findings highlight the regulatory roles of differential NUP expression in modulating the progenitor differentiation process. Similar to the previous findings that the NPC numbers remain relatively constant in aging[44], we found that keratinocyte differentiation is not associated with NPC number reduction, despite the drastic downregulation of most NUPs. Our results also suggest that the fractions of off-pore NUPs, localizing in the cytoplasmic and nuclear compartments, are likely to play key roles in modulating the differentiation process.

In this paper, we focused on investigating the downregulation of NUP93, as it is the most downregulated NUP at the mRNA level in primary human keratinocyte differentiation. Although NUP93 was reported as one of the most stable proteins in rat brain tissue[44], we found that NUP93 protein level was significantly reduced in human keratinocyte differentiation. NUP93 knockdown in keratinocytes also effectively reduced NUP93 protein expression. Furthermore, we found that NUP93's incorporation into the NPCs is moderately reduced in differentiation, suggesting that NUP93's incorporation to the nuclear pores is not static in this process[45,46]. These data indicate that NUP93 proteins in human keratinocytes are subjected to active turnover, and they are much more unstable as compared to the observations from rat brain tissues. In addition to the nuclear-pore

localization, we found that NUP93 can localize to the cytoplasm as well as nucleoplasm. Based on our quantification of the western blotting data, the off-pore NUP93 fractions are likely to be the major contributing factors of NUP93 reduction in differentiation. In particular, since NUP93 chromatin-binding has been reported in Drosophila S2 cells and in the colorectal cancer cell line DLD-1[45,47], it will be interesting in the future to clarify if NUP93 can bind to chromatin in keratinocytes and if NUP93 binding reduces in differentiation.

Although NUP93 depletion was found to impair NPC assembly in vitro[46], NUP93 reduction in keratinocyte differentiation or with knockdown did not affect NPC number based on our STORM quantification. NUP93 depletion was also associated with altered nuclear pore permeability or nuclear import in the literature[33,34,48]. However, our reporter assays did not detect drastic differences in protein transport with NUP93 knockdown or in the differentiation process. As the knockdown strategy was designed to reduce but not deplete NUP93 in the progenitor-state keratinocytes, the remaining NUP93 protein levels were likely to be sufficient for maintaining the basic function of the nuclear pores.

With NUP93 knockdown in keratinocytes, we identified the nuclear translocation of NF-κB p65/p50 transcription factors. Although NF-κB activation is involved in inflammation and immune responses in other cell types such as macrophages and T cells, NK-κB activation promotes differentiation in keratinocytes. The nuclear translocation of p50 was reported in the upper differentiated layers of skin epidermis, and the overexpression of p65 or p50 was demonstrated to drive keratinocyte growth arrest[35,49]. Blockage of p65/p50 nuclear translocation had been reported in human cutaneous squamous cell carcinoma clinical samples, synergizing with the Ras oncogene to drive carcinogenesis[50]. Thus, p65/p50 nuclear translocation is essential for keratinocyte terminal differentiation, and the nuclear translocation of p65/p50 is involved in differentiation induction with NUP93 knockdown. In addition, this work suggests that NUP93 reduction could be linked to ER stress to indirectly drive the activation of the NF-κB pathway. Although NUP210 was previously connected to ER stress[5], NUP210 is not expressed in keratinocytes. NUP98 and RAE1 were recently identified in regulating keratinocyte differentiation[3], our analyses of the RNA-seq data suggest that NUP93 is uniquely involved in suppressing the immune genes in progenitor-state keratinocytes.

Our findings also shed light on the differences between primary human cells versus cancer cell lines, in the context of NUP93's functions. NUP93 overexpression in metastatic breast cancer lines was reported to enhance the nuclear import of SMAD through interaction with importin[51], although this NUP93-importin interaction was not observed in a different study using hepatocellular carcinoma cells[52]. In this study, we find that NUP93 knockdown does not drastically influence the subcellular localization of SMAD in primary human keratinocytes. Furthermore, it is the reduction of NUP93 (instead of overexpression) that

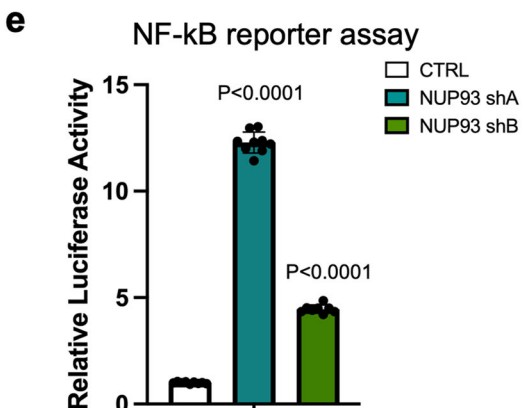

**Fig. 7 NUP93 knockdown in the progenitor state induces p65/p50 nuclear translocation. a–d** Representative images of p65/p50 immunostaining and quantification of the relative ratio of fluorescence in the nucleus versus cytoplasm in each cell (N:C ratio), comparing NUP93 knockdown versus control. (p65: CTRL vs NUP93-shA $p = $ <0.0001, CTRL vs NUP93-shB $p = 0.0123$; p50: CTRL vs NUP93-shA $p = $ <0.0001, CTRL vs NUP93-shB $p = 0.0363$. $n = 30$ per condition, one-way ANOVA with post-hoc test). Data are represented as mean ± standard deviation. **e** Luciferase assay comparing NF-κB reporter activity in keratinocytes between control knockdown versus NUP93 knockdown. One-way ANOVA with post-hoc test, $N = 9$ biological replicates. Data are represented as mean ± standard deviation.

enhances p65/p50 nuclear import. These different findings indicate that NUP93's action in gene regulation is highly context-dependent, especially between the normal non-transformed cells and metastatic cancer cells.

Altogether, we have identified that primary human keratinocyte differentiation involves reduction in NUP expression, but not in nuclear pore numbers. Our findings suggest that the off-pore fraction of NUPs contribute to their differential expression levels between the progenitor state and the differentiation state. We further identified that the high expression of NUP93 is linked to the suppression of NF-κB activation in epidermal progenitor maintenance. Building on the conceptual and technical foundation established in this study, future work characterizing the NUPs will enrich our understanding of their diverse roles in gene regulation in both normal tissue homeostasis and disease pathogenesis.

## Methods

**Primary cell culture.** This study using primary human keratinocytes was reviewed by Northwestern University IRB (Institutional Review Board) and was determined as not human research. Surgically discarded neonatal foreskin was obtained from Northwestern SBDRC (Skin Biology & Diseases Resource Based Center), and tissue collection was approved by IRB (#STU00009443). Primary keratinocytes isolated from 6-7 different de-identified donors were pooled for all the experiments associated with this study. To culture and maintain the keratinocytes in the undifferentiated condition, a 50:50 mix of two culture media, KSFM (Gibco, #17005-142) and Medium 154 (Gibco #M-154-500), was used. For calcium-induced differentiation, the keratinocytes were seeded in confluency with the addition of 1.2 mM $CaCl_2$ for 4 days. For comparing keratinocyte gene expression in low, sub, or full confluence, 0.3 million keratinocytes were seeded to a 10 cm plate (low confluency), a 33 mm plate (sub confluency) or a well in a 24-well plate (full confluency) for collection on the next day. Images were taken the next day to verify the confluency, before RNA extraction and qRT-PCR.

**Plasmid construction.** The oligo sequences of the shRNAs targeting NUP93 were designed using the BLOCK-iT RNAi Designer (ThermoFisher, oligo sequences included in Supplementary Data 2). The oligos were annealed and cloned into the pLKO.1 puro vector (Addgene #8453). Oligo synthesis was performed by Integrated DNA Technologies (IDT). The mCherry and H2B-GFP constructs were cloned into the pLZRS expression vector. The mCherry-NES construct was cloned into pLZRS using the PKI NES sequence (LALKLAGLDI)[53]. The mCherry-LacZ-NLS construct was cloned into pLZRS using the NLS sequence (PKKKRKV)[54].

**Gene transfer and expression in primary keratinocytes.** Phoenix cells or HEK293T cells were cultured in DMEM (Gibco, high glucose, pyruvate) supplemented with 10% FBS. Transfection was performed using Lipofectamine 3000 (ThermoFisher). Viral supernatant was collected 2–3 days post transfection. For infecting the keratinocytes, the viral supernatant was added together with 20 μg/mL polybrene, with centrifugation ($310 \times g$, 60 min at 32 °C). Puromycin selection was performed by adding 2 μg/mL Puromycin to keratinocytes 2 days post viral infection for 48 h.

**Western blotting.** The protein concentration of keratinocyte lysate was measured by BioRad Bradford Protein Assay, and 10–15 ug μg of protein per sample was loaded for SDS-PAGE.

The separated proteins were transferred onto a (polyvinylidene difluoride) PVDF membrane. After blocking using the Odyssey Blocker PBS (LI-COR) + 2.5% BSA for 1 h at the room temperature, primary antibodies were added to the PVDF membrane for overnight incubation at 4 °C. The primary antibodies used in this study include NUP205 (H1) (Santa Cruz sc-377047, 1:200), NUP93 (E-8) (Santa Cruz sc-374399, 1:200), NUP133 (Santa Cruz sc-376699, 1:200), lamin A/C (ThermoFisher MA5-35284, 1:1000; or Santa Cruz sc-376248, 1:1000). Secondary antibody (Goat anti-mouse LI-COR 96-68020, Goat anti-mouse LI-COR 96-32211) incubation was performed at room temperature for 1 h at 1:15000 dilution.

**Quantitative real-time PCR.** RNA was extracted from the keratinocytes using the Quick-RNA MiniPrep Kit (Zymo Research). Reverse transcription was performed using the SuperScript VILO master mix (Invitrogen). Real-time PCR reactions were performed using the PowerUp SYBR Green Master Mix (Thermo-Fisher) with the QuantStudio3 system (ThermoFisher). Technical triplicates were performed for each sample, and 18 S ribosomal RNA was used for normalization. For statistical analysis, unpaired $t$-test was performed using GraphPad Prism. Oligo sequences for the primers used in this study are included in Supplementary Data 2.

**Nuclei extraction, fixation, and staining for STORM imaging.** Keratinocytes were resuspended in 200 uL of nuclei extraction buffer (10 mM HEPES pH 7.4, 1.5 mM $MgCl_2$, 10 mM KCl, Protease inhibitor without EDTA (Roche)) per million of cells. An equal volume of nuclei extraction buffer + 0.4% NP-40 was added to the cell suspension, followed by incubation on ice for 2 min. The suspension was quickly spun to pellet nuclei, and nuclei extraction buffer was added again for 5 min on ice. Nuclei were pelleted and resuspended in PBS. Nuclei were immobilized onto poly-L-lysine treated coverslips by centrifuging for 5 min at $450 \times g$. Nuclei were then fixed using 10% formalin for 10 min at room temperature. After fixation, nuclei were permeabilized and blocked in PBS with 2.5% normal goat serum and 0.5% Triton X-100 for 30 min at room temperature. After permeabilization and blocking, coverslips were incubated with appropriate primary antibodies (Mab414: Biolegend MMS-120P at 1:50, NUP133: Sigma HPA059767 at 1:100, NUP93: Santa Cruz sc-374399 at 1:50) at 4 °C for 16 h. Alexa Fluor 647-Secondary antibodies conjugates were prepared using NHS-coupling reaction. A Donkey anti-mouse full-length IgG and a Donkey-antirabbit IgG-F(ab')₂ (Jackson ImmunoResearch) were prepared using the same chemical reaction. The antibody conjugates were purified using Amicon Ultra-0.5 Centrifugal Filter Unit with a molecular size cut-off of 50 kDa. The degree of label for the antibodies were determined to be ~1 dye per antibody using Nanodrop Absorption 2000 Spectrometer.

**Keratinocyte immunofluorescence staining.** Keratinocytes growing on Polylysine-coated glass coverslips were fixed by 10% formalin at room temperature for 10 min. After permeabilization (PBS with 0.4% Triton X-100) and blocking (2.5% normal goat serum), primary antibodies were added to the coverslips overnight at 4 °C. The primary antibodies used in this study include p50 (E-10) (Santa Cruz, sc-8414, 1:50), p65 (Cell Signaling, #8242, 1:100), IKKα (B-8) (Santa Cruz, sc-7606, 1:50), PDI (Cell Signaling, #3501, 1:100), and SMAD2/3 (Santa Cruz, sc-133098, 1:50). Secondary antibodies (Life Technologies, A11304 and A11005) were diluted at 1:400 for incubation at room temperature for 1 h. DNA was stained using Hoechst 33342 at 1ug/mL.

**Progenitor competition assay**. $4 \times 10^5$ keratinocytes expressing GFP were mixed with $4 \times 10^5$ keratinocytes expressing DsRed to regenerate epidermal tissue, by seeding onto a $1 cm^2$ piece of human dermis raised to the air-liquid interface. The GFP-expressing keratinocytes were transduced with non-targeting control shRNA. The DsRed-expressing keratinocytes were transduced with one of the following shRNAs in each regeneration experiment: non-targeting control shRNA, NUP93 shA, or NUP93 shB. 4 mL FAD medium (provided by SDBRC at Northwestern) was used for the first 2 days, and E medium (provided by SDBRC at Northwestern) was used for the following 4 days, with daily medium change. The tissue was fixed after 6 days of tissue regeneration using 10% formalin (Fisher).

**STORM imaging system setup and procedure**. Each well in the silicone-spacer chamber was filled up with ~100 μL imaging buffer for immediate STORM imaging. This imaging buffer contains 50 mM Tris (pH = 8.0), 10 mM NaCl, 0.5 mg/mL glucose oxidase (Sigma, G2133), 2000 U/mL catalase (Sigma, C30), 10% (w/v) D-glucose, and 100 mM cysteamine. Another piece of cover glass was sealed to the chamber with gentle press to cover the chamber from the top. This covered chamber was then mounted onto the microscope stage for imaging acquisition.

For STORM imaging acquisition, a custom-built STORM imaging system was used with a Nikon Ti-2E fluorescence microscope and using a 100x TIRF objective lens (CFI Apochromat HP 100x). The details of the imaging setup were described previously[55]. A 647-nm continuous-wave laser was used to excite the fluorophore with a power density of ~2 kW cm-2. The incident angle of the excitation beam was set to be slightly larger than the TIRF angle. The exposure time was 10 ms and 20,000 frames were acquired for each of the STORM imaging movies for imaging reconstruction.

**Confocal laser scanning microscopy (CLSM)**. For live cell imaging, keratinocytes were growing in Clear-Bottom Black 96-well or 12-well plate (Corning). CLSM imaging was performed for keratinocytes in keratinocyte culture media. A Leica SP5 confocal microscope was used to acquire the CLSM images. A 488-nm and a 561-nm laser line was used to image the GFP and mCherry channels, respectively. For capturing images from immunofluorescent staining, a 63x oil objective was used. Images were captured in a format of $1024 \times 1024$ at 600 Hz. Sequential scans were used for the 488 nm and 561 nm laser to acquire images in green and red channels.

**Fluorescence recovery after photobleaching (FRAP)**. For the FRAP experiments, keratinocytes with both GFP and mCherry signals were located and a FRAP series was performed. Briefly, a 40x objective was used and a region of $\sim 5 \times 5 \mu m^2$ was selected for photobleaching. This region was bleached for 8 s using 100%-intensity of 561-nm laser with a zoom-in mode of the FRAP Wizard of Leica SP5 confocal imaging acquisition program. Then a post-FRAP series was acquired every 10 s to around 500 s.

For FRAP video analyses, the boundary of each nucleus was outlined using the GFP-H2B signal in the GFP channel, and a $\sim 5 \times 5 \mu m^2$ region-of-interest within the nucleus was selected to measure the average signal change over the time course of FRAP movie.

**RNA-seq library construction and sequencing**. RNA was extracted from keratinocytes using Quick-RNA MiniPrep Kit (Zymo), and mRNA was subsequently purified from the RNA extraction using the Poly(A) mRNA Magnetic Isolation Module (New England Labs). RNA-seq library was constructed using the NEBNext® Ultra™ II Directional RNA Library Prep Kit for Illumina. Libraries were sequenced on HiSeq4000 using 50 base pair single-end reads by Northwestern University NUseq Core.

**Clonogenicity assay**. The day prior to seeding keratinocytes, mouse fibroblast 3T3 cells were treated with 15 ug/mL mitomycin C for 2 h in DMEM and seeded onto 6-well plates at ~60% confluency. DMEM media was changed to FAD media 1 h before seeding keratinocytes. 1000 keratinocytes treated with control shRNA or knockdown shRNA were seeded onto the 3T3 cells in triplicates. FAD media was changed every 2–3 days for 10–12 days. Once the control shRNA colonies reached a size >1 mm² the media was removed and the 3T3 cells were washed away using PBS. The remaining colonies were fixed with cold 1:1 methanol:acetone solution for 5 min. The fixation solution was removed, and plates were left to air dry for 5 min. Colonies were then stained with crystal violet stain for 10 min and rinsed with water. Colonies >1 mm² were counted for each condition.

**Apoptosis assay**. Keratinocytes infected with either control or NUP93 shRNA were seeded onto a 24-well plate. Hydrogen peroxide (2 mM) was added to control knockdown keratinocytes for 6 h at 37 °C as a positive control. The Mitoview 633 (Biotium, 70055 T) reagent was reconstituted in DMSO according to manufacturer's instructions and added at 1:1,000 and incubated for 20 min in the $CO_2$ incubator. Prior to imaging, the DNA stain Hoechst was added at 10 ug/mL for 5 min.

**Luciferase reporter assay**. The pHAGE NFkB-TA-LUC-UBC-GFP-W plasmid from addgene (#49343) was used to make lentivirus. Keratinocytes were infected with this virus and either control or NUP93 shRNA. The Dual-Glo(R) Luciferase Assay System from Promega (E2920) was used to lyse cells and the luciferase luminescence signals were measured. The luminescence was normalized by cell number.

**ER stress induction by brefeldin A**. Keratinocytes were treated with brefeldin A (B6542-5MG, sigma) dissolved in DMSO at 2 ug/mL final concentration for 24 h. DMSO treatment alone was used as controls for the imaging and qRT-PCR experiments.

**Cell fractionation for protein quantification using western blotting**. Nuclei from undifferentiated and differentiated keratinocytes were extracted using the same procedures as in nuclei extraction for STORM imaging above, and the supernatant containing the cytoplasmic proteins was saved. Pelleted nuclei were directly lysed in urea lysis buffer. The protein in the supernatant containing the cytoplasmic fraction was precipitated with acetone at −20 for 24 h. The precipitated protein was pelleted and resuspended in an equal volume of urea lysis buffer as the nuclei pellet.

**Statistical information**

*STORM imaging data analysis*. The single-molecule localization (SML) step was performed using the ThunderSTORM plugin of ImageJ. Before imaging reconstruction, a cross-correlation-based drift correction was applied, and out-of-focus single-molecules (sigma > 200 nm) were rejected.

Further cluster analyses were carried out in MATLAB or Python. Briefly, a coordinate list containing all the single-molecule localizations belonging to a single nucleus were used to perform HDBSCAN clustering of individual NPC. The selection of minimal point per cluster in HDBSCAN was optimized using Monte-Carlo simulation to provide high-fidelity clustering of each NPC. The SML density and NPC density within the entire nucleus was calculated by dividing all the localization events with the nuclear area. The number of single-molecule localizations

within each NPC and the NPC diameter were extracted from each cluster in HDBSCAN. Student's *T*-test was used for statistical analysis comparing UD vs DF.

*Fluorescence recovery after photobleaching (FRAP) data analysis.* The fluorescence recovery traces were fitted using an exponential association function using GraphPad Prism 9 to extract the half-time for the fluorescence recovery process. The diffusion coefficients were calculated following Axelrod et al., 1976.

*RNA-seq data analysis.* Fastq files were aligned to the human genome using HISAT2. Alignments were filtered and sorted using SAMtools. Htseq was used to build counts tables. DESeq2 was used for determining differentially expressed genes, with >2 fold change than both controls and $p < 0.05$. Heatmaps was plotted using ggplots in Rstudio or Prism. DAVID was used for Gene Ontology analysis.

*Western blotting quantification.* The western blots were scanned using the LI-COR Odyssey Clx system, and the fluorescent intensity was quantified using the LI-COR Image Studio Software. The intensity from Lamin A from the same lane was used as a loading control to compare the protein expression levels among different conditions.

*Nuclear to cytoplasmic (N:C) ratio quantification.* The images were analyzed using ImageJ. The boundary of each nucleus was outlined using the signals of H2B-GFP (live cells) or DAPI (fixed and stained cells), and the boundary of the entire cell was outlined using the signal from mCherry reporters (live cells) or the IF signal from the transcription regulators (fixed and stained cells). The cytoplasmic signal was calculated by subtracting the nuclear signal from the signal within the whole cells. The N:C ratio was then calculated by dividing the nuclear signal by the cytoplasmic signal in each cell. *T*-test was used to calculate p-values.

**Statistics and reproducibility**. For nuclear pore number quantification using STORM images, 5–15 images of nuclei from each condition were quantified and the Student's *t* test was used to compare each pair of two conditions. For FRAP data analysis, 6–9 FRAPs were performed for each condition. Statistical analyses For FRAP experiments were performed for Half Time for Recovery using one-way ANOVA with post-hoc test. For Nuclear to Cytoplasmic (N:C) ratio quantification, 15–30 independent measurements were used for each condition, and the student's *t* test was used for the comparisons between two conditions. All the RT-qPCR experiments were performed at least 3 times, and one-way ANOVA with post-hoc tests were used for statistical analyses. All the western blotting experiments were performed at least 3 times. For shRNA mediated knockdown, 4 independent shRNAs were designed and validated for NUP93. For RNA-seq data analysis, two different controls (non-targeting shRNA, shRNA targeting GFP) were used to compare with four NUP93 knockdown conditions (using 4 different shRNAs).

**Reporting summary**. Further information on research design is available in the Nature Portfolio Reporting Summary linked to this article.

## Data availability
The RNA-seq data have been deposited to GEO as GSE209655. In addition to the RNA-seq data generated in this study, previously generated RNA-seq data sets used in this paper are also available from GEO (GSE150799, GSE127223). The super-resolution imaging data are available by request (yang.zhangfl@gmail.com), and the data processing codes are available on FOIL Github. The uncropped images for western blotting are included as Supplementary Fig. 8. Source data are included as Supplementary Data 3.

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

## Acknowledgements

We appreciate the scientific discussions with Drs. Jason Brickner, Alec Wang, and Curt Horvath about this project. We thank the collegiality from other Bao Lab members during the progression of the project. We thank the confocal microscopy acquisition services from the Biological Imaging Facility (BIF) and Center of Advanced Molecular Imaging (CAMI), tissue acquisition from the Skin Biology & Diseases Resource-Based Center (SBDRC), and the next-generation sequencing services by NUseq core at Northwestern University. This work is supported by the following funding sources: National Institutes of Health grant R01 AR075015 (X.B.), American Cancer Society Research Scholar Grant RSG-21-018-01-DDC (X.B.), National Institutes of Health grant R01GM140478 (C.S., H.F.Z., X.B.), National Science Foundation grants CBET-1706642, CHE-1954430, EFRI-1830969 (H.F.Z.), National Institutes of Health grants R21GM141675 and R01GM143397 (Y.Z. and H.F.Z.), National Institutes of Health grants R01GM139151, U54CA268084 (H.F.Z.), Illinois Society for the Prevention of Blindness Standard Research Grant (Y.Z., H.F.Z.), Office of Biological and Environmental Research of the U.S. Department of Energy Atmospheric System Research Program Interagency Agreement grant DE-SC0000001, a Pilot & Feasibility Award from the Skin Biology and Diseases Resource-Based Center (SBDRC, AF057216) at Northwestern (X.B.), Searle Leadership fund (X.B.), and the Northwestern COVID Recovery Fund (X.B.).

## Author contributions

X.B., Y.Z., A.N., H.F.Z., and C.S. designed the project and the experiments; A.N., Y.Z., L.B. and X.B. performed the experiments and analyzed the data; Y.Z., A.N., and X.B. participated in data visualization and illustration; H.M. and B.B. helped with super-resolution data acquisition and data analysis. X.B., C.S., H.F.Z., and Y.Z. obtained funding to support this project; X.B., Y.Z., A.N., and H.F.Z. wrote the paper.

## Competing interests

The authors declare no competing interests.
