## [Peer Review File · Communications Biology]

Referee expertise:

Referee #1: NPC, Nups

Referee #2: nuclear structure and function, Nup93

Referee #3: NPC, Nups

Reviewers' comments:

Reviewer #1 (Remarks to the Author):

Neely et al describe in their study "Reduction of Inner-Ring Nucleoporin Promotes Differentiation via Enhancing Nucleocytoplasmic Trafficking and NF- κ B Activation" that during in vitro keratinocyte differentiation gene expression of nucleoporins, the proteins forming the nuclear pore complexes in the nuclear envelope, decreases as uncovered by RNAseq. This is accompanied by a reduction of protein levels of some nucleoporins without affecting NPC numbers. The authors go on presenting data which they interpret as showing that downregulation of one of the nucleoporins, Nup93, diminishes progenitor regenerative capacity, accelerates nuclear transport and de-repressed differentiation programs.

It is known that NPC composition can be different between cell lines (PMID: 23511206) and especially change during developmental processes, most prominent examples being probably GP210/POM210 (PMID: 22264802) in muscle cell differentiation and Nup133 neural stem/progenitor cell differentiation (PMID: 18539113). In the current manuscript RNAseq data indicate that RNA levels of a variety of nucleoporins are changed during in vitro keratinocyte differentiation. It is unclear why the authors then focus on Nup93 and whether the effects seen are due to a specific change in Nup93 or rather reflect global changes on NPC structure and function. E.g. a reduction in clone numbers upon NUP93 knockdown (Fig. 4c,d) might be simply due to decreased cell viability given that Nup93 is an essential gene. Importantly, the statements on enhanced nucleocytoplasmic trafficking are invalid: The assays used in Fig. 5 does not reflect active nuclear transport but rather leakage of an inert substrate. Fig. 7 says something about steady state localization of one specific substrate and but cannot address kinetics. Fig. S3 indicates that nuclear export is functional but does not show kinetics.

Specific points:

- 1.) Line 127: Can the authors explain the rationale behind addressing Nup133 even though it does not seem to be the most affected of the coat nucleoporins? There are other coat nucleoporins that have been connected to cell proliferation/differentiation alterations such as Nup107 and Seh1.
- 2.) Line 129 "All these three NUPs showed progressive downregulation at both the mRNA and the protein levels in the differentiation time course". One might argue that what is seen here for 30 different nucleoporins is a change of the total protein levels in cells rather than at the nuclear envelope. In the light of the analysis shown in fig 1a-c, how can the authors be sure that this change is specific to the nuclear pore complexes rather than the whole cell?
- 3.) Line 183: Please note that MAB414 does not recognize Nup107 and Nup155 as these nucleoporins do not contain FG-repeats. Also Fig 2I needs to be accordingly corrected.
- 4.) Line 156 "we started by using a NUP133 antibody": It would have been useful here to use a nucleoporin from your analysis in Fig 1b/c which shows minimal change between the progenitor and differentiated states. This way the reader can be sure that what is seen is not caused by technical differences between the two samples and reflect the fold alterations seen in figure 1C.
- 5.) Line 236 "The expression of NUP93-shRNA, both not 237 control shRNA, strongly reduced the representation...": not clear what meant.
- 6.) Line 269 "These data suggest that NUP93 reduction in keratinocytes promotes the switch from the progenitor state towards differentiation, by derepressing differentiation genes.": it is not clear whether this is specific for Nup93 or a general NPC defect.

- 7.) Line 311: "In the differentiation state, reduced NUP93 incorporation to the NPCs accelerated the nucleocytoplasmic transport kinetics and enriched p65/p50 nuclear localization": As NES-mCherry reporter is not affected as shown in Figure S3 by Nup93 downregulation this is an overstatement. Also no reporters with nuclear localization signals have been investigated. It would be better to be more specific and point to the nucleocytoplasmic distribution of p65 and p50 in this case.
- 8.) Fig 1a/2b/2l/3a: NPC scheme is misleading: The inner ring (green) is positioned at the place where the central channel nucleoporins are located. The coat nucleoporins do not cover the entire pore membrane, Rather inner ring nucleoporins establish contact to the transmembrane ring. The authors might check expert reviews of the NPC field.
- 9.) Fig 1g,h,i/Fig 4b/Fig S1c: molecular size markers are lacking.
- 10.) Fig 1g,h,i/Fig 4b: lamin A/C or lamin B?
- 11.) Fig 5: I would estimate half times of recovery of about 70 sec for panel B and of about 130 sec for panel D. Given that both measurements include control (treated) keratinocytes this discrepancy needs to be explained and the interpretation of the differences seen within a experiments between control versus treated or undifferentiated versus differentiated reevaluated.

Reviewer #2 (Remarks to the Author):

Major Comments:

The manuscript under review entitled "Reduction of Inner-Ring Nucleoporin Promotes Differentiation via Enhancing Nucleocytoplasmic Trafficking and NF- κ B Activation" show interesting structural and functional aspects of nucleoporins, however, the mechanistic significance of the involvement of Nup93 in this process needs more robust substantiation and corroboration.

1. While it is remarkable to detect transcript level changes in the Nups during differentiation of keratinocytes, do the Nups show transcript level changes upon altered confluency of cells i.e from low, sub-confluent to confluent cells?
2. Since Nup93 interacts with Nup188 and with Nup205, is there a transcriptional feedback between these factors before and after differentiation? or is the transcript level decrease independent of one another in this sub-complex?
3. Presumably the downregulation in the Nup levels has been quantified from across multiple independent biological replicates? since the lamin loading control shows some variability in the representative blots.
4. Is the downregulation of Nups also detected for both the on-pore and off-pore nups at the mRNA, protein and numbers of on and off pore nups?
5. Based on the numbers data is it reasonable to surmise that Nup93 subunits are in a dynamic equilibrium between the nuclear pore and nucleoplasm? if this were to be the case, are the numbers of different nups interdependent on one another and does this point to a functional significance in the context of keratinocyte differentiation?
6. Is the decreased incorporation of the nups during differentiation a consequence of altered nuclear size?
7. Are the numbers of MAB414 altered upon any of the Nup knockdowns?
8. Fig.5b & d: This is an interesting data. Would be useful to plot and include the statistical significance of the Nup93 Kd compared to the non-targeting control in this figure. Is the nuclear import rescued upon over-expression of wild type Nup93?
9. Nup93 knockdown de-repressed differentiation gene expression: While Nup93 Kd reveals a significant transcriptional de-repression, can this effect be attributed to increased nuclear import of various transcription factors upon Nup93 Kd or is it a consequence of Nup93 depletion per se? can these two seemingly connected events be uncoupled from one another?
10. Fig.6a: Is there a significant difference in the gene expression profiles in shA from that of shB, as

Fig.6A shows a striking difference in the heat map profiles between shA and shB?

11. Figs 6c-f: Is the increase in transcript levels a consequence of Nup93 knockdown or increased N/C transport?

12. It is also intriguing as to how GO categories that are specific to keratinocyte differentiation are enriched selectively upon Nup93 Kd? Can this effect be rescued upon Nup93 overexpression? Is there a mechanistic basis for the same?

13. Fig.7: Similarly, can the effect of p50/p65 translocation be rescued? alternatively, what is the consequence of p50/p65 overexpression in the context of Nup93, or Nup133 knockdown for instance?

14. Fig.7; While it is exciting to note that Nup93 Kd drives the nuclear translocation of the transcription factors p50/p65, it would be useful to perform a ChIP-PCR assay to demonstrate that p50/p65 are indeed enriched on the promoters of genes required for keratinocyte differentiation.

15. More importantly, what is the mechanistic basis of Nup93 regulated differentiation? is it possible to uncouple nuclear transport from that of Nup93 Kd mediated gene expression changes?

Minor edits

1. Fig.1g-i: Please specify the lamin sub-type

2. Fig.1g: Is the altered levels of lamins statistically significant?

3. line: 141: change to though

4. line#236: Change to but

5. line#418: correct to lipofectamine

6. Fig#5b: Please label X-axis

Reviewer #3 (Remarks to the Author):

Neely and coworkers report that expression of multiple nuclear pore proteins (nucleoporins; nups) is reduced at mRNA and protein level during differentiation from progenitors to keratinocytes. The authors focus on NUP93 and NUP205 that are most strongly reduced and also NUP133. Using STORM imaging, the authors find that the levels of NUP133 at nuclear pore complexes (NPCs) as well as NPC density are stable despite the reduction in cellular NUP133 concentration. In contrasts, NPCs of terminally differentiated cells contain less NUP93 compared to progenitor cells. To explore a possible role of NUP93 in progenitor cells, shRNAs are used to deplete NUP93. This causes an increase in nuclear accumulation of p50 and p65 and a reduction in regenerative capacity. The authors also argue that nucleocytoplasmic transport kinetics are altered.

The manuscript is well written, the data presented are of good quality and the conclusions are convincing with a few exceptions:

The conclusion from FRAP experiments that NUP93 depletion affects nucleocytoplasmic transport should be revised. I have two concerns: 1) The differences between CTRLsh and NUP93KD in Figure 5 look very minor and based on a relative low number of cells. Are these differences significant? The authors should extract recovery parameters and compare control vs shRNA by statistical means. 2) The term "nucleocytoplasmic trafficking" refers to regulated and active transport of macromolecules across NPCs. Probably increased permeability rather than (active) transport. Depletion on the NUP93/Nic96 complex in yeast, worms and flies has been shown to increase passive transport (reduced permeability barrier; see ref 32 + PMID: 10831607 + PMID: 20547758). Based on these previous observations, it is likely that effects observed here are caused by an increase in passive transport across the nuclear envelope.

The images provided in Figure 7 (and to some degree Figures S2-S3) indicate that the cell density is lower when NUP93 is depleted. Does this reflect that these cells are not dividing (as expected from previous studies) and if so, how does this affect the interpretation of p50 and p65 localisation?

Minor points:

Line 74: Ref #8 is not related to with NPC structure.

Line 102: Change "(at 25nm)" to "(at 25 nm resolution)".

Line 183: To my understanding, MAB414 recognizes primarily NUP62, NUP153, NUP214 and NUP358.

Line 215: The title of this section is not adequate: the section does not include data on nucleocytoplasmic transport.

Point-by-Point Response to Reviewers

(Original comments in gray, responses in black, new data in figures in Blue.)

Reviewer #1 (Remarks to the Author):

Neely et al describe in their study “Reduction of Inner-Ring Nucleoporin Promotes Differentiation via Enhancing Nucleocytoplasmic Trafficking and NF- κ B Activation” that during in vitro keratinocyte differentiation gene expression of nucleoporins, the proteins forming the nuclear pore complexes in the nuclear envelope, decreases as uncovered by RNAseq. This is accompanied by a reduction of protein levels of some nucleoporins without affecting NPC numbers. The authors go on presenting data which they interpret as showing that downregulation of one of the nucleoporins, Nup93, diminishes progenitor regenerative capacity, accelerates nuclear transport and de-repressed differentiation programs.

It is known that NPC composition can be different between cell lines (PMID: 23511206) and especially change during developmental processes, most prominent examples being probably GP210/POM210 (PMID: 22264802) in muscle cell differentiation and Nup133 neural stem/progenitor cell differentiation (PMID: 18539113). In the current manuscript RNAseq data indicate that RNA levels of a variety of nucleoporins are changed during in vitro keratinocyte differentiation. It is unclear why the authors then focus on Nup93 and whether the effects seen are due to a specific change in Nup93 or rather reflect global changes on NPC structure and function. E.g. a reduction in clone numbers upon NUP93 knockdown (Fig. 4c,d) might be simply due to decreased cell viability given that Nup93 is an essential gene. Importantly, the statements on enhanced nucleocytoplasmic trafficking are invalid: The assays used in Fig. 5 does not reflect active nuclear transport but rather leakage of an inert substrate. Fig. 7 says something about steady state localization of one specific substrate and but cannot address kinetics. Fig. S3 indicates that nuclear export is functional but does not show kinetics.

We appreciate the comments and suggestions from the reviewer. The Ori 2013 paper (PMID: 23511206) and the Lupu 2008 paper (PMID: 18539113) are very interesting. The roles of NUP210, on the other hand, appear to be more complicated. – After the initial paper (PMID:22264802), the same group published another paper (PMID:25778917) three years later, describing that NUP210 influences differentiation independent from NPC but associates with ER stress. We have updated the manuscript to include all these references.

Among all the NUPs that are drastically downregulated in keratinocyte differentiation, we decided to focus on NUP93 mainly because NUP93 is the most downregulated NUP in keratinocyte differentiation at the mRNA level, and this downregulation was further validated at the protein level by western blotting. We have further clarified this in the manuscript. As different NUPs are likely to influence gene expression through diverse mechanisms, we envision characterizing additional NUPs in skin biology as the next steps.

To address whether the effects of NUP93 knockdown could be associated with global changes of NPC structure and function, we performed several experiments. *First,*

apoptosis assay showed that NUP93 knockdown in keratinocytes did not induce apoptosis (new data included as Supplementary Fig. 4d). The knockdown efficiency of the shRNAs is about 75-85%, with 15-25% remaining NUP93 protein in the keratinocytes. This level of knockdown efficiency is comparable to the NUP93 reduction in the differentiation process (~25% remaining NUP93 protein as compared to the progenitor state). It is likely that the remaining NUP93 protein, after knockdown, was sufficient to support cell viability. Second, we quantified NPC numbers in NUP93 knockdown and did not observe significant changes (new data included as Fig. 5a-e).

We apologize for using the term “enhanced nucleocytoplasmic trafficking” in the previous version. We have corrected this throughout the manuscript, including the title. We used several reporter assays to exam both the nuclear pore permeability (for insert protein using mCherry as a reporter) and the active transport function (using NLS-LacZ-mCherry and mCherry-NES as reporters) and did not detect significant differences in keratinocytes with NUP93 reduction (new data included as Supplementary Fig. 3). Thus, the basic protein transport functions of the nuclear pores are largely intact in the progenitor-state keratinocytes with NUP93 knockdown.

Specific points:

1.) Line 127: Can the authors explain the rationale behind addressing Nup133 even though it does not seem to be the most affected of the coat nucleoporins? There are other coat nucleoporins that have been connected to cell proliferation/differentiation alterations such as Nup107 and Seh1.

We apologize for the confusion. We started from NUP133 as a marker for quantifying nuclear pore numbers because of its characteristic ring structure previously established using super-resolution imaging (PMID: 23845946) and the commercial availability of a high-quality antibody that works for endogenous protein. NUP133 helped us to establish the technical foundation of the super-resolution imaging capacity. Building on the observations from NUP133, we further validated the quantification using the mAB414 antibody that recognizes multiple NUPs with FG repeats.

We appreciated that the reviewer pointed out NUP107 and Seh1. NUP107 is downregulated in keratinocyte differentiation (-1.881016), although not as much as NUP93 (-2.62). Seh1, on the other hand, is not drastically downregulated in keratinocyte differentiation (-0.3141276). We learned that different NUPs are likely to function through different pathways in gene regulation (PMID: 37353594) and we are interested in further exploring the roles of different NUPs including NUP107 and Seh1 for future studies.

2.) Line 129 ” All these three NUPs showed progressive downregulation at both the mRNA and the protein levels in the differentiation time course”. One might argue that what is seen here for 30 different nucleoporins is a change of the total protein levels in cells rather than at the nuclear envelope. In the light of the analysis shown in fig 1a-c, how can the authors be sure that this change is specific to the nuclear pore complexes rather than the whole cell?

We agree with the reviewer that the total protein level is not identical to the levels at the nuclear envelope. Indeed, this was why we applied the super-resolution technique

to evaluate the NUP levels at the nuclear pores. In agreement with the prediction from the reviewer, our cell fraction data suggest that both NUP133 and NUP93 can localize other subcellular compartments in addition to the nuclear envelope (new data included as Fig. 3j-o).

3.) Line 183: Please note that MAB414 does not recognize Nup107 and Nup155 as these nucleoporins do not contain FG-repeats. Also Fig 2l needs to be accordingly corrected.

We appreciate this suggestion, and we have corrected this in the manuscript. MAB414 recognizes NUP62 (central channel), NUP153 (nuclear basket), NUP214 and NUP358 (cytoplasmic).

4.) Line 156 “we started by using a NUP133 antibody”: It would have been useful here to use a nucleoporin from your analysis in Fig 1b/c which shows minimal change between the progenitor and differentiated states. This way the reader can be sure that what is seen is not caused by technical differences between the two samples and reflect the fold alterations seen in figure 1C.

We agree that it would be a better idea to start with a different antibody in this context. We moved the MAB414 (recognizes multiple NUPs) imaging of undifferentiated and differentiated keratinocytes before NUP133.

5.) Line 236” The expression of NUP93-shRNA, both not 237 control shRNA, strongly reduced the representation...”: not clear what meant.

We apologize for the typo. It should have been “but” instead of “both” in this sentence. This is now corrected.

6.) Line 269 “These data suggest that NUP93 reduction in keratinocytes promotes the switch from the progenitor state towards differentiation, by derepressing differentiation genes.”: it is not clear whether this is specific for Nup93 or a general NPC defect.

The 4 shRNAs we generated for NUP93 knockdown resulted in 15-25% remaining protein level in keratinocytes cultured in the progenitor-state condition. In the differentiation process, NUP93 is downregulated to ~23% as compared to the progenitor state, based on our quantification using western blotting. Therefore, the remaining level with knockdown is comparable to the level observed in differentiation.

To determine if NUP93 knockdown may have caused NPC defect, we further performed several experiments. Leveraging super-resolution imaging, we quantified NPC numbers using MAB414 in keratinocytes with NUP93 knockdown versus control and observed no significant changes (new data included as Fig. 5a-e). We evaluated the protein transport function of NPCs with NUP93 using NLS or NES reporters and did not find drastic differences as compared to control knockdown (new data included as supplementary Fig. 3 & 4). These data suggest that the remaining NUP93 protein with knockdown was sufficient to retain the NPC functions in these keratinocytes.

7.) Line 311: “In the differentiation state, reduced NUP93 incorporation to the NPCs accelerated the nucleocytoplasmic transport kinetics and enriched p65/p50 nuclear localization”: As NES-mCherry reporter is not affected as shown in Figure S3 by Nup93

downregulation this is an overstatement. Also no reporters with nuclear localization signals have been investigated. It would be better to be more specific and point to the nucleocytoplasmic distribution of p65 and p50 in this case.

We thank the reviewer for pointing this out. We generated an NLS-LacZ-mCherry reporter and tested it in differentiation as well as in NUP93 knockdown. No significant differences were observed (new data included as supplementary Fig. 3).

8.) Fig 1a/2b/2l/3a: NPC scheme is misleading: The inner ring (green) is positioned at the place where the central channel nucleoporins are located. The coat nucleoporins do not cover the entire pore membrane, Rather inner ring nucleoporins establish contact to the transmembrane ring. The authors might check expert reviews of the NPC field.

We apologize for the confusion. The NPC scheme is now updated and incorporated into the figures.

9.) Fig 1g,h,i/Fig 4b/Fig S1c: molecular size markers are lacking.

We thank the reviewer for pointing this out. The molecular size markers are now included.

10.) Fig 1g,h,i/Fig 4b: lamin A/C or lamin B?

It is lamin A/C. We have now clarified this in the figure legends and have updated the labels in the figures.

11.) Fig 5: I would estimate half times of recovery of about 70 sec for panel B and of about 130 sec for panel D. Given that both measurements include control (treated) keratinocytes this discrepancy needs to be explained and the interpretation of the differences seen within a experiments between control versus treated or undifferentiated versus differentiated reevaluated.

We apologize for the confusion. The differences between the previous panel B vs panel D were due to the different objective lenses used for the two sets of experiments. We have now repeated the FRAP experiments with consistent microscope settings, and the figures are updated.

Reviewer #2 (Remarks to the Author):

Major Comments:

The manuscript under review entitled "Reduction of Inner-Ring Nucleoporin Promotes Differentiation via Enhancing Nucleocytoplasmic Trafficking and NF- κ B Activation" show interesting structural and functional aspects of nucleoporins, however, the mechanistic significance of the involvement of Nup93 in this process needs more robust substantiation and corroboration.

We appreciate the constructive comments from the reviewer, and we have substantially revised the manuscript accordingly. The details addressing individual points are listed below.

1. While it is remarkable to detect transcript level changes in the Nups during differentiation of keratinocytes, do the Nups show transcript level changes upon altered confluency of cells i.e from low, sub-confluent to confluent cells?

This is a great suggestion, as keratinocytes' commitment to differentiation is known to be influenced by culture density. Early differentiation marker gene keratin 1 (KRT1) was reported to be strongly induced as keratinocytes switch from sub-confluence to full confluence (PMID:7530273).

To determine if keratinocyte density influences the expression of NUPs, we seeded 0.3 million cells to a 10-cm plate (low), a 33mm plate (sub), or in a well of a 12 well plate (full). We were able to recapitulate the induction of KRT1 as previously reported. Interestingly, we found that the expression of representative NUPs, including NUP133, NUP93 and NUP205, were all significantly downregulated just with full confluence (new data included as supplementary Fig. 1), suggesting that NUP downregulation is an early event in keratinocyte differentiation.

2. Since Nup93 interacts with Nup188 and with Nup205, is there a transcriptional feedback between these factors before and after differentiation? or is the transcript level decrease independent of one another in this sub-complex?

We appreciate this question. According to our RNA-seq data of NUP93 knockdown, NUP205 is slightly downregulated (average log2 fold change: -0.75, p=0.02), and NUP188 is also slightly downregulated (average log2 fold change: -0.5, p=0.07) with NUP93 knockdown. These 2 NUPs did not pass our statistical filter and fold change cut off (p<0.05, fold change >2). Other than NUP188 and NUP205, NUP37 was also downregulated (average log2 fold change: -1.19, p=0.018), suggesting that NUP93 downregulation can affect other NUPs in addition to NUP188 and NUP205. Not all NUPs are strongly downregulated, such as the nuclear basket NUPs (NUP98, NUP153, and NUP50). These data suggest that there is a potential transcriptional feedback among a specific subset of NUPs. A heatmap showing the expression of different NUPs is now included as Supplementary Fig. 5a, and the fold changes of different NUPs are now included as a separate tab in Supplementary Data 1.

3. Presumably the downregulation in the Nup levels has been quantified from across multiple independent biological replicates? since the lamin loading control shows some variability in the representative blots.

Yes, we confirm that all western blots shown have been performed with 3 or more biological replicates. These replicates are reflected in the quantification and statistical analyses.

4. Is the downregulation of Nups also detected for both the on-pore and off-pore nups at the mRNA, protein and numbers of on and off pore nups?

We agree with the reviewer that it is important to examine both the "on-pore" and "off-pore" fractions of NUPs. The "on-pore" fraction was examined using super-resolution

imaging and quantification. To investigate the “off-pore” NUPs, we perform cell fractionation and used western blotting to quantify the NUPs in the cytoplasmic and nuclear fractions. Interestingly, we observed a cytoplasmic fraction of NUP133 and NUP93 enriched in the undifferentiated state. Furthermore, the nuclear fraction showed a stronger reduction in western blotting as compared to what we observed on the nuclear envelope (new data included as Fig. 3j-o). These findings suggest that the off-pore fraction of NUP is more differentially regulated in the process of keratinocyte differentiation.

5. Based on the numbers data is it reasonable to surmise that Nup93 subunits are in a dynamic equilibrium between the nuclear pore and nucleoplasm? if this were to be the case, are the numbers of different nups interdependent on one another and does this point to a functional significance in the context of keratinocyte differentiation?

We agree that NUP93's subcellular localization is likely to be dynamic, based on the data comparing the undifferentiated and differentiated keratinocytes. The total protein level of NUP93 is reduced in differentiation and the fraction of NUP93 associated with the nuclear pores is also reduced in differentiation based on the super-resolution imaging and quantification.

With NUP93 knockdown or with differentiation, we did not observe significant reduction of nuclear pore numbers (new data included as Fig. 5a-e). The expression of a subset of other NUPs was mildly reduced with NUP93 knockdown (new data included as Supplementary Fig. S5a). Suggesting that the potential inter-dependence of expression may not fully explain the drastic downregulation of the NUPs in keratinocyte differentiation process.

6. Is the decreased incorporation of the nups during differentiation a consequence of altered nuclear size?

We examined the size of extracted nuclei, based on their area size labelled by NUP133 or mAb414, and did not observe significant differences (Supplementary Fig. 2). In addition, we only observed decreased incorporation of NUP93, but not NUP133 and mAB414. Thus, the specifically decreased incorporation of NUP93 was unlikely to be a consequence of altered nuclear size.

7. Are the numbers of MAB414 altered upon any of the Nup knockdowns?

To address this, we performed super-resolution imaging of MAB414 comparing NUP93 knockdown versus non-targeting control, in primary keratinocytes cultured in the progenitor-state condition. There was no statistical difference among the non-targeting control versus NUP93 knockdown using different shRNAs (new data included as Fig. 5a-e).

8. Fig.5b & d: This is an interesting data. Would be useful to plot and include the statistical significance of the Nup93 Kd compared to the non-targeting control in this figure. Is the nuclear import rescued upon over-expression of wild type Nup93?

We appreciate this suggestion. We repeated the mCherry FRAP experiment for the mCherry reporter, with increased sample sizes. Statistical analyses were performed using half-time, and the data were not statistically significant. These data suggest that

NPC's function allowing inert proteins to diffuse is not significantly altered with NUP93 knockdown or keratinocyte differentiation.

We overexpressed NUP93 in the differentiated state to see if it was sufficient to stop differentiation from occurring. However, we found that NUP93 overexpression alone does not drastically change the expression of differentiation genes such as *GRHL3*, *OVOL1*, and *KLF4* (new data included as Supplementary Fig. 5d). We also compared the subcellular localization of p65 and p50 in keratinocytes with NUP93 overexpression versus control and did not detect a drastic difference (new data included as Supplementary Fig. 6b).

9. Nup93 knockdown de-repressed differentiation gene expression: While Nup93 Kd reveals a significant transcriptional de-repression, can this effect be attributed to increased nuclear import of various transcription factors upon Nup93 Kd or is it a consequence of Nup93 depletion per se? can these two seemingly connected events be uncoupled from one another?

This is a great question. NUP93 knockdown in primary human keratinocytes appeared to specifically alter the subcellular localization of the NFkB transcription factors p65 and p50, but not IKKA or SMAD2/3. We further constructed an NLS-LacZ-mCherry reporter to test the active import process in keratinocytes with NUP93 knockdown versus control, and this also showed no significant difference (new data included as Supplementary Fig. 3). Thus, these data suggest that NUP93 reduction likely has a specific effect on p65/p50 translocation rather than altering the basic transport function.

10. Fig.6a: Is there a significant difference in the gene expression profiles in shA from that of shB, as Fig.6A shows a striking difference in the heat map profiles between shA and shB?

We apologize for the confusion. The two shRNAs did not have the same knockdown efficiency, which at least partially explains the differences between the gene expression profiles. We have included two additional shRNAs to achieve a total of four shRNAs in the revision process, quantified their knockdown efficiency and repeated the RNA-seq experiment, and we still observe a comparable trend with similar GO terms (new data included in Fig. 6).

11. Figs 6c-f: Is the increase in transcript levels a consequence of Nup93 knockdown or increased N/C transport?

We believe the induction of differentiation/immune marker genes is a consequence of NUP93 knockdown. Our data using multiple reporters suggest that the changes in N/C transport are not statistically significant.

12. It is also intriguing as to how GO categories that are specific to keratinocyte differentiation are enriched selectively upon Nup93 Kd? Can this effect be rescued upon Nup93 overexpression? Is there a mechanistic basis for the same?

We appreciate this suggestion. We generated an overexpression construct, achieving about 6-fold increase of NUP93 protein levels in keratinocytes. Interestingly, NUP93 overexpression alone was not sufficient to significantly alter the level of differentiation

marker genes (new data included as Supplementary Fig. 5d), suggesting that additional regulators might be involved in the regulation of differentiation gene expression together with NUP93.

13. Fig.7: Similarly, can the effect of p50/p65 translocation be rescued? alternatively, what is the consequence of p50/p65 overexpression in the context of Nup93, or Nup133 knockdown for instance?

The overexpression of NUP93 also did not alter the translocation of p50/p65 (new data included as Supplementary Fig. 6b), suggesting that the NUP93 level was necessary but not sufficient in modulating this process.

14. Fig.7; While it is exciting to note that Nup93 Kd drives the nuclear translocation of the transcription factors p50/p65, it would be useful to perform a ChIP-PCR assay to demonstrate that p50/p65 are indeed enriched on the promoters of genes required for keratinocyte differentiation.

We attempted p65 ChIP-seq multiple times with NUP93 knockdown. As NUP93 knockdown led to cell cycle exit, keratinocyte barely grew after knockdown. It was technically challenging to obtain sufficient cell numbers to obtain reliable ChIP data. To circumvent this technical issue, we leveraged a luciferase-based NFkB reporter that includes 4 tandem copies of NFkB DNA binding consensus. NUP93 knockdown resulted in significant increase of the luciferase activity, suggesting that p65/p50 DNA binding is indeed increased in keratinocytes with NUP93 knockdown (new data as Fig. 7e).

15. More importantly, what is the mechanistic basis of Nup93 regulated differentiation? is it possible to uncouple nuclear transport from that of Nup93 Kd mediated gene expression changes?

We appreciate this great question. Based on the data from the reporter assays, we found that the nuclear pore permeability and its basic function in controlling NLS/NES transport is not drastically changed with NUP93 knockdown. The NF-kB transcription factors p50/p65, but not all proteins that can translocate between the cytoplasm and the nucleus, were uniquely enriched in the nucleus. Thus, this nuclear localization of p50/p65 is likely to be independent from the basic transport function of the nuclear pores, in keratinocytes with NUP93 knockdown. In addition, we noticed that the reporter proteins were often accumulated right outside the nucleus in keratinocytes with NUP93 knockdown (Supplementary Fig. 7a,b). Protein accumulation in the ER has been previously linked to NF-kB activation (PMID: 7781611). Keratinocytes treated with Brefeldin A, which induces protein aggregation in ER, also showed nuclear translocation of p50/p65 as well as induction of NF-kB target genes (Supplementary Fig. 7c-f), similar to NUP93 knockdown. Both NF-kB and ER stress have been recently linked to keratinocyte differentiation induction (PMID: 30890691, PMID: 14724177, PMID: 12695650, PMID: 12673201, PMID: 12571598). Furthermore, we identified upregulation genes known to be induced by ER stress in both NUP93 and in keratinocyte differentiation. These findings suggest that NUP93 downregulation is linked to the stress response, which induces NF-kB nuclear translocation to promote keratinocyte differentiation.

Minor edits

1. Fig.1g-i: Please specify the lamin sub-type

Thank you for pointing this out. We have now updated the lamin sub-type in the figures and in the legends. Lamin A/C was used in this study.

2. Fig.1g: Is the altered levels of lamins statistically significant?

Based on our RNA-seq data, lamin A/C is not significantly altered in keratinocyte differentiation. We apologize that the lamin appears to be uneven in this blot. The relative expression of NUP93 at the protein level was quantified relative to the lamin level within the same lane.

3. line: 141: change to though

We apologize that the sentence was hard to read. This sentence is now revised to "...by removing cytoplasmic proteins that can contribute to non-specific antibody binding".

4. line#236: Change to but

We apologize for the typo. It should have been "but" instead of "both" in this sentence. This is now corrected.

5. line#418: correct to lipofectamine

Thank you for catching the typo. It is now corrected.

6. Fig#5b: Please label X-axis

We have now labelled the x-axis (time, sec) for all the FRAP data.

Reviewer #3 (Remarks to the Author):

Neely and coworkers report that expression of multiple nuclear pore proteins (nucleoporins; nups) is reduced at mRNA and protein level during differentiation from progenitors to keratinocytes. The authors focus on NUP93 and NUP205 that are most strongly reduced and also NUP133. Using STORM imaging, the authors find that the levels of NUP133 at nuclear pore complexes (NPCs) as well as NPC density are stable despite the reduction in cellular NUP133 concentration. In contrasts, NPCs of terminally differentiated cells contain less NUP93 compared to progenitor cells. To explore a possible role of NUP93 in progenitor cells, shRNAs are used to deplete NUP93. This causes an increase in nuclear accumulation of p50 and p65 and a reduction in regenerative capacity. The authors also argue that nucleocytoplasmic transport kinetics are altered.

The manuscript is well written, the data presented are of good quality and the conclusions are convincing with a few exceptions:

The conclusion from FRAP experiments that NUP93 depletion affects nucleocytoplasmic transport should be revised. I have two concerns:

- 1) The differences between CTRLsh and NUP93KD in Figure 5 look very minor and based on a relative low number of cells. Are these differences significant? The

authors should extract recovery parameters and compare control vs shRNA by statistical means.

We appreciate the suggestion from the reviewer. We repeated these experiments with increased number of cells and calculated the halftime from each FRAP experiment. The differences were not statistically significant. We have adjusted our conclusions accordingly in the manuscript (new data added as Fig. 5).

- 2) The term “nucleocytoplasmic trafficking” refers to regulated and active transport of macromolecules across NPCs. Probably increased permeability rather than (active) transport. Depletion on the NUP93/Nic96 complex in yeast, worms and flies has been shown to increase passive transport (reduced permeability barrier; see ref 32 + PMID: 10831607 + PMID: 20547758). Based on these previous observations, it is likely that effects observed here are caused by an increase in passive transport across the nuclear envelope.

We thank the reviewer for pointing this out. We agree that “permeability” is more suitable in this context of the mCherry reporter, and we have corrected accordingly in the manuscript, and added the two other references to the paper. Although NUP93 depletion (implying 0% remaining proteins) reduced NPC permeability barrier in the literature, NUP93 knockdown in keratinocytes (with 15-25% remaining proteins) was not sufficient to significantly affect permeability. In addition, we constructed an NLS-LacZ-mCherry reporter to evaluate active nuclear import, and did not observe a significant difference. One explanation is that the remaining NUP93 protein levels were sufficient to retain the permeability, with the knockdown (instead of KO) approach.

The images provided in Figure 7 (and to some degree Figures S2-S3) indicate that the cell density is lower when NUP93 is depleted. Does this reflect that these cells are not dividing (as expected from previous studies) and if so, how does this affect the interpretation of p50 and p65 localisation?

We apologize for the confusion. NUP93 knockdown impairs proliferation, which accounts for the lower cell density on the image. We have included new data, with undifferentiated keratinocytes seeded at low versus higher confluency and stained for p65 and p50. We found that cell confluency alone does not drastically alter the subcellular localization of p50 and p65 (new data included as Supplementary Fig. 6a).

Minor points:

Line 74: Ref #8 is not related to with NPC structure.

Thank you for catching this, and this reference is now removed.

Line 102: Change “(at 25nm)” to “(at 25 nm resolution)”.

Thanks for the suggestion. It is now corrected.

Line 183: To my understanding, MAB414 recognizes primarily NUP62, NUP153, NUP214 and NUP358.

We appreciate the reviewer for pointing this out, and we have corrected this the manuscript accordingly.

Line 215: The title of this section is not adequate: the section does not include data on nucleocytoplasmic transport.

We appreciate this suggestion. We agree that the previous title was not ideal, and the title is now updated to better reflect our findings.

Reviewers' comments:

Reviewer #3 (Remarks to the Author):

While the authors have addressed most of my queries, I still have a few additional queries on the role of nucleoporins in differentiation.

1. Fig.1 g-i: If Nup numbers remain unaltered, what is the implication of the decrease in Nup protein levels? does the decrease in total protein levels suggest a decrease in the relative numbers of Nups? Does this result counter the data from the STORM imaging data? What are the possible mechanisms that counter (buffer) these differences if any?

2. Fig: Data from PMID: 29791854 and other related papers suggests that Nup133 is part of the Y-complex - a crucial complex in regulating the formation of the NPC. I would suggest revisiting the relative spatial locations for better schematic representations of each of the Nup subunits, which would be useful for the reader.

3. Fig.3: Although this aspect has been discussed to some extent in the results, I still think that the relative differences in the total protein levels and relative localization of Nup93 in the cytoplasmic or nuclear fractions could be clarified better (both from existing literature and insights from this work), considering the added complexity of its decrease during keratinocyte differentiation? In other words, where is the maximum sub-population of Nup93 associated with the cytoplasmic or Nup (outer/inner) during differentiation? and how does this impact differentiation? This scenario is also consistent with Nup133.

4. Fig.4: Do the high levels of Nup93 in the progenitor populations, correlate with relatively higher levels of pluripotency marks as well?

5. Is there an overlap or crosstalk between genes deregulated upon Nup93 and Nup133 knockdowns?

Reviewer #4 (Remarks to the Author):

The authors have performed multiple experiments to address the reviewers' comments. I am satisfied with their responses to specific points and I recommend the manuscript for publication.

(Reviewer's original comments are in Gray, responses are in Black, additions of new data are indicated in Blue).

1. Fig.1 g-i: If Nup numbers remain unaltered, what is the implication of the decrease in Nup protein levels? does the decrease in total protein levels suggest a decrease in the relative numbers of Nups? Does this result counter the data from the STORM imaging data? What are the possible mechanisms that counter (buffer) these differences if any?

As shown in Fig. 3j-o, we performed cell fractionation (into the nuclear and cytoplasmic fractions) and used western blotting to investigate the subcellular distribution of NUP133 and NUP93. Using this approach, we identified that NUP93 and NUP133 both have significantly higher cytoplasmic enrichment in the progenitor state as compared to the differentiation state. In the case of NUP93, but not NUP133, we also observed significant reduction in the nuclear fraction in differentiation.

To further clarify this, we performed immunostaining and compared NUP133/NUP93 localization with PDI (Protein Disulfide Isomerase, a marker for the cytoplasm) in progenitor-state keratinocytes using con-focal microscopy. In agreement with our western blotting data, our immune-fluorescent staining data showed clear cytoplasmic localization of NUP133 and NUP93. NUP93's nucleoplasmic localization is also visible from these images. (New data included as supplementary Fig. 2c,d.)

These findings complement the STORM imaging data, which only focused on the nuclear envelope. Taken together, our findings suggest that the NUPs can localize to the cytoplasm and the nucleoplasm, in addition to the well-recognized location of the nuclear envelope.

2. Fig: Data from PMID: 29791854 and other related papers suggests that Nup133 is part of the Y-complex - a crucial complex in regulating the formation of the NPC. I would suggest revisiting the relative spatial locations for better schematic representations of each of the Nup subunits, which would be useful for the reader.

We agree with the reviewer that it would be a good idea to mention the Y complex in the manuscript. We've made the following changes to further clarify this. *First*, we have emphasized that NUP133 is part of the Y complex in the manuscript and have cited the paper pointed out by the reviewer. *Second*, we clarified that the Y complex is also part of the "coat" NUPs as labelled in the illustration. *Third*, we realized that the previous version of the illustration was not sufficient to clarify that NUP133, as part of the Y complex, localizes to both the nuclear and cytoplasmic sides of the nuclear pore (Bley et al, Science 2022; Mosalaganti et al., Science 2022; Petrovic et al., Science 2022). We have updated Fig.2I accordingly to reflect that NUP133 localizes to both sides.

3. Fig.3: Although this aspect has been discussed to some extent in the results, I still think that the relative differences in the total protein levels and relative localization of Nup93 in the cytoplasmic or nuclear fractions could be clarified better (both from existing literature and insights from this work), considering the added complexity of its decrease during keratinocyte differentiation? In other words, where is the maximum sub-population of Nup93 associated with the cytoplasmic or Nup (outer/inner) during differentiation? and how does this impact differentiation? This scenario is also consistent with Nup133.

There are great questions. We have expanded the Discussion Section to address the reduction of NUP93. Although the reduction of NUP93 at the nuclear pores is statistically significant, the fold change is moderate. On the other hand, we observe drastic reduction in nuclear and cytoplasmic fractions. We speculate that the cytoplasmic fraction and a chromatin binding fraction of NUP93 can contribute to differentiation for the following two reasons. *First*, a couple of recent papers identified that NUP93 can bind to chromatin. Future work verifying and clarifying NUP93's chromatin binding in keratinocytes will be informative. *Second*, our work suggests that ER stress could mediate the NF- κ B activation upon NUP93 knockdown. It is possible that the cytoplasmic localization of NUP93 is involved in suppressing ER stress. In the case of NUP133, we have identified a cytoplasmic pool that is drastically reduced in differentiation (Fig 3. m-o). We now understand that different NUPs can engage in diverse cellular processes, especially when they are localized to other cellular compartments. Although the functional assays of NUP133 were not within the scope of this paper, future work characterizing the detailed functions of NUP133 will be interesting to understand the significance of NUP133 cytoplasmic localization.

4. Fig.4: Do the high levels of Nup93 in the progenitor populations, correlate with relatively higher levels of pluripotency marks as well?

This is a very interesting question, as progenitors (uni-potent adult stem cells) have unique characteristics as compared to embryonic stem cells or iPS cells. KLF4, one of the classic "yamanaka factors" (pluripotency markers), actually upregulates in differentiation and functions as a driver for the termination differentiation process (Segre et al., Nature Genetics, 1999). In addition to keratinocytes, KLF4 also drives monocyte differentiation (Feinberg et al., the EMBO Journal, 2007). Both OCT4 and SOX2 are expressed at very low levels in keratinocytes. Therefore, the class pluripotency markers for ES cells or iPS cells are not well correlated with the progenitor populations of human keratinocytes. In agreement with NUP93's role in suppressing differentiation in the progenitor state, our RNA-seq data indicate that NUP93 knockdown also significantly upregulated KLF4 (a differentiation activator in keratinocytes). SOX2 and OCT4 are not among the significantly altered genes with NUP93 knockdown.

5. Is there an overlap or crosstalk between genes deregulated upon Nup93 and Nup133 knockdowns?

The reviewer raised an excellent question, regarding whether the differentially expressed genes with NUP93 knockdown can be recapitulated by knocking down other NUPs. To address this, we compared NUP93 knockdown with our recently published knockdown data of two other NUPs, NUP98 and RAE1 (new data included as Supplementary Fig. 5 e-j). NUP93 knockdown uniquely upregulate immune related genes; however, the 3 NUPs do share 172 upregulated genes and 208 downregulated genes, with GO terms related to the proliferation and differentiation processes. NUP93 knockdown does not drastically influence the expression of NUP133 (average Log2 fold change is only -0.1998), suggesting that NUP133 is not controlled by NUP93. Taken together, these findings suggest that the immune related genes are uniquely repressed by NUP93, although different NUP knockdown can alter a subset of shared differentiation and proliferation related genes. NUP93 knockdown does not strongly

influence the expression of NUP133. NUP133 knockdown is outside the scope of this study, as NUP133 was only used as a marker in this study for quantifying nuclear pore numbers using SMLM.